# PROMPT OPTIMIZATION WITH LOGGED BANDIT DATA

## ABSTRACT

We study how to use naturally available user feedback, such as clicks, to optimize large language model (LLM) pipelines for generating personalized sentences using prompts. Naive approaches, which estimate the policy gradient in the prompt space, suffer either from variance caused by the large action space of prompts or bias caused by inaccurate reward predictions. To circumvent these challenges, we propose *Direct Sentence Off-policy gradient* (DSO), which estimates the policy gradient by leveraging similarity among generated sentences, substantially reducing variance while suppressing the bias. Empirical results on our newly established suite of benchmarks, called **OfflinePrompts**, demonstrate the effectiveness of the proposed approach in generating personalized descriptions for movie recommendations, particularly when the number of candidate prompts is large.

## 1 INTRODUCTION

As more systems with large language model (LLM)-generated text are starting to become operational, we are naturally collecting increasing amounts of logged user feedback from their system interactions. These feedback signals provide valuable information on whether the prompt or generated sentence was effective for the user. Unlike conventional datasets used for LLM training (Stiennon et al., 2020), this feedback is available for all users at little cost, providing opportunities for personalizing sentence generation in applications like search, recommendations, and educational chatbots. Thus, it is worth developing a method to use such naturally logged user feedback to enhance the quality and outcome (i.e., reward) of language generation.

To optimize sentence generation, we focus on learning a *prompt policy* (i.e., which prompt to use for a particular user or situation). As detailed in the following, learning a prompt policy is attractive for reasons of (1) safety, (2) cost, and (3) accessibility. First, for most applications it is a key requirement to not produce harmful outputs (Bai et al., 2022). By only adapting the prompts without fine-tuning the LLM itself, we do not run the risk of removing the safety properties of the underlying LLM. Second, compared to the LLM itself, a prompt policy can be a small model, which reduces the required computational resources and the amount of data needed for training (Deng et al., 2022). Additionally, compared to using a hand-engineered prompt, a prompt policy enables automated prompt optimization and promises greater personalization. Third, a prompt policy can be trained even in situations where the LLM is closed-weights and available only through an inference API, which makes prompt-policy learning feasible even for small companies or individuals.

While learning a prompt policy is attractive as argued above, using logged user feedback and performing *off-policy learning* (OPL) of a new prompt policy entails several challenges due to the *partial* nature of the feedback. Specifically, the logged data is bandit feedback, containing the reward for only the action (prompt) chosen by the *logging* policy (i.e., the one used in past operations) and not for the other actions that a new policy may choose. This data-generating process is outlined in Figure 1: for each coming user, the logging policy chooses which prompt to use for generating the sentence; then, each user observes only the sentence generated by the chosen prompt and thus reveals the reward (e.g., click) for only this sentence. A naive way to deal with such counterfactuals is to regress the reward and use imputed rewards instead (Stiennon et al., 2020; Jaques et al., 2017; Snell et al., 2022b). However, imputation is often not accurate enough under covariate shift (Swaminathan & Joachims, 2015) and complex relations between prompts and rewards. An alternative is importance sampling, which re-weighs reward observations w.r.t. the ratio of prompt distribution between the logging and target policies. Nonetheless, this approach suffers from severe variance when the action space is large (Saito et al., 2024) and bias when the logging policy does not fully explore the action

Figure 1: **Overview of the prompt-based sentence personalization with logged bandit feedback**. For each coming user, a policy chooses which prompt to use to generate sentences with a frozen LLM. Each user observes only the sentence generated by the chosen prompt and provides the reward for the corresponding sentence. Logged bandit feedback is partial in that we cannot observe rewards for the sentences generated by prompts *not* chosen by the logging policy. Examples are generated by ChatGPT-3.5 (Brown et al., 2020).

space (Sachdeva et al., 2020). These challenges can be particularly problematic in our language generation setting, where we need to deal with a rich and diverse set of candidate prompts as actions.

The key shortcoming of the standard approaches lies in treating each prompt independently, not taking the information about the generated sentence into account. In response, **this paper explores and presents a method to leverage the similarity among generated sentences to make large-scale OPL for prompt-guided language generation efficient and tractable**. Specifically, our *Direct Sentence Off-policy gradient* (DSO) estimates the policy gradient in the sentence space (i.e., not in the action space of prompts) to take the generated sentence into account. We enable this by applying the importance weight in the (marginalized) sentence space and by re-sampling the action conditioned on the sentence when calculating the score function. Because the former effort contributes to reducing the scale of the importance weight and the latter works as an implicit data augmentation about the prompt, we can expect variance reduction of DSO compared to typical OPL methods. Moreover, by aggregating similar prompts and sentences using kernels, DSO also keeps the bias small.

Finally, we develop and conduct experiments on a newly-developed OPL benchmark suite, called **OfflinePrompts**, in both synthetic and full-LLM settings. The results demonstrate the effectiveness of our approach, particularly when the number of prompts is large. We provide our benchmark suite, including the full-LLM environment for generating personalized movie descriptions for recommendations based on the MovieLens (Harper & Konstan, 2015) dataset, as an open-source resource. Its easy-to-use API will accelerate both future research and practical applications of OPL of prompt-guided language generation with logged bandit feedback.

Our contributions are summarized as follows.

- **Problem formulation**: We present a view of prompt-policy learning from naturally available feedback as OPL of contextual bandits, providing a pathway for advancing OPL for prompts.
- **Tailoring OPL methods for language generation**: We identify how to effectively leverage similarity in generated sentences by taking the gradient directly in the sentence space.
- **Benchmarks and open-source**: We conduct extensive experiments and provide a new benchmark suite as open-source software for future research and applications.

## 2 RELATED WORK

This section summarizes notable related work. Extended discussion can be found in Appendix A.

**Prompt Tuning.** Prompt tuning is a cost-efficient approach for optimizing language generation for some specific downstream applications, including translation, summarization, and sentiment analysis. Unlike fine-tuning, which requires the updates of the millions of parameters of the LLM itself, prompt

tuning reuses a "frozen" pre-trained LLM and optimizes only the choice of the special tokens added to the original input sentence called *prompts* (Brown et al., 2020). Deng et al. (2022) presented a reinforcement learning (RL) formulation of prompt tuning, which optimizes the prompts via policy gradient by treating a frozen LLM as a black-box reward generator. While this formulation is relevant to ours, the critical limitation of Deng et al. (2022) and similar online exploration papers (Dwaracherla et al., 2024) is to assume that feedback (i.e., reward) is easily accessible. Unfortunately, such an assumption is unrealistic in many real-world applications where online interactions with users can be costly, harmful, or sometimes even unethical (Matsushima et al., 2021; Gilotte et al., 2018). Instead, we present a way to leverage logged user feedback naturally collected through past operations.

**Reinforcement Learning for Language Generation.** RL from Human Feedback (RLHF) is a widely-studied approach to align the output of LLMs using human annotation (Christiano et al., 2017; Stiennon et al., 2020; Ouyang et al., 2022; Lin et al., 2024). Specifically, RLHF asks human annotators to compare two sentences and provide labels to indicate which sentence is more appropriate for a downstream task (e.g., translation). Then, using the pairwise feedback, RLHF trains a model to predict the task-specific score of each sentence to preserve the preference. The key challenges of RLHF are in two folds: (1) RLHF incurs substantial cost and ethical concerns for human annotation (Bai et al., 2022; Lee et al., 2023) and (2) monitoring if annotators provide sufficiently reliable labels for RLHF, as done in Stiennon et al. (2020); Ouyang et al. (2022), can be difficult when preferred sentences change among annotators in tasks related to personalization. Our approach of learning a contextual prompt policy using logged bandit feedback naturally resolves the above difficulties of RLHF.

**Off-Policy Evaluation and Learning.** Off-Policy Evaluation and Learning (OPE/OPL) studies how to use naturally collected user feedback to evaluate and learn new contextual bandit or RL policies (Saito et al., 2021; Fu et al., 2020). Regression-based and importance sampling (IS)-based approaches are prevalent in OPL. First, the regression-based approach trains a reward predictor and then optimizes a new policy using imputed rewards. While this approach performs well when the reward predictor is accurate for the entire action (i.e., prompt) space, such an accurate regression is often demanding due to the issues such as counterfactuals and covariate shift (Swaminathan & Joachims, 2015). In contrast, the IS-based approach aims to estimate the policy gradient unbiasedly from actually observed rewards by correcting the distribution shift (Precup et al., 2000). However, IS often suffers from high variance and deficient support, particularly when the action space is large (Saito & Joachims, 2022; Saito et al., 2023; Sachdeva et al., 2024). To overcome the limitations of naive approaches, Saito et al. (2024) has recently proposed a two-stage OPL framework called POTEC, which first chooses which cluster among pre-defined action-clusters to use by applying cluster-wise IS and then chooses which action within the chosen cluster to use. However, good clusterings are often hard to identify, and this approach discards information about the generated sentences. In response, we present a way to leverage similarity among sentences by estimating the policy gradient directly in the sentence space. Another related literature is Kallus & Zhou (2018), which discuss OPE of deterministic policies in a continuous action space. While we share the ideas of using kernels with Kallus & Zhou (2018), our idea comes from a different notion of deriving the gradient directly in the (marginalized) sentence space. Moreover, the theoretical analysis is entirely different, as we apply kernels to the logging policy, while Kallus & Zhou (2018) do not. Finally, our new benchmark, which simulates the personalized generation of sentences, is a unique contribution of ours to the OPE/OPL community.

## 3 PROBLEM FORMULATION

We start by formulating prompt optimization as a new type of OPL problem, which we call *contextual bandits with auxiliary outputs*.

Let $u \in \mathcal{U} \subseteq \mathbb{R}^{d_u}$ be a $d_u$-dimensional user feature vector (e.g., demographic profile or user id), sampled from an unknown distribution $p(u)$. Let $q \in \mathcal{Q} \subseteq \mathbb{R}^{d_q}$ be a *query* (e.g., query to a frozen LLM), sampled from a conditional distribution $p(q|u)$. Let $a \in \mathcal{A}$ be a (discrete) *prompt*, where each prompt is associated with some vectorial embedding, $e_a \in \mathbb{R}^{d_e}$, where $d_e$ is the dimension of the embeddings. The prompt is used to generate a sentence via a frozen LLM. This process can be formulated as a procedure of sampling sentence $s \in \mathcal{S}$ as an auxiliary output from the stochastic output distribution of the LLM: $p_{\text{LLM}}(s|q, a)$. A user will respond to the output sentence and provide

some reward $r \in \mathbb{R}$ (e.g., click or purchase), where $r$ follows $p(r|u,q,s)$. Let $\pi \in \Pi$ be a *prompt policy* where $\pi(a|u,q)$ is the probability of choosing *prompt $a$* for *context $x := (u,q) \in \mathcal{X}$*. Our goal is to optimize the prompt policy to maximize the expected reward, defined as

$$V(\pi) := \mathbb{E}_{\underbrace{p(u)p(q|u)}_{=p(u,q)} \pi(a|u,q) \underbrace{p_{\text{LLM}}(s|q,a)p(r|u,q,s)}_{=p(r,s|u,q,a)}}[r] = \mathbb{E}_{p(x)\pi(a|x)p(r,s|x,a)}[r].$$

When running a prompt policy $\pi_0 (\neq \pi)$ as part of an operational system, it works as a *logging* policy and generates logged feedback of the following form:

$$\mathcal{D} := \{x_i, a_i, s_i, r_i\}_{i=1}^n \sim \prod_{i=1}^n p(x)\pi_0(a|x)p_{\text{LLM}}(s|x,a)p(r|x,s)$$

where $n$ is the data size and $i$ is its index. The logged data informs us whether the prompt ($a_i$) results in a high reward or not ($r_i$) for a particular context ($x_i$). However, a difficult aspect of using the logged data is that the reward observation is *partial*, i.e., it is observed only for the prompt chosen by the logging policy ($\pi_0$) but not for all the other actions. This can be particularly challenging when training a new policy $\pi$ on the logged data, as $\pi$ may choose actions that are not chosen by $\pi_0$. Thus, we need to address such *counterfactuals* and *distribution shift* between the logging and learning policies when using logged data for a reliable policy optimization (Swaminathan & Joachims, 2015).

In the rest of the paper, we parameterize the policy as $\pi_\theta$ using some parameters $\theta \in \Theta$ (e.g., a neural network). We also define $q(x,a) := \mathbb{E}[r|x,a]$ and $q(x,s) := \mathbb{E}[r|x,s]$. Finally, $z \sim p(z)$ indicates that we sample a single random variable $z$ from the probability distribution $p(\cdot)$, for any random variable $z$ and its corresponding probability distribution.

### 3.1 CONVENTIONAL APPROACHES

We first review direct applications of typical OPL methods and discuss their limitations.

**Regression (Konda & Tsitsiklis, 1999).** A typical way of using logged data is to train a reward predictor $\hat{q}$ (Stiennon et al., 2020; Jaques et al., 2017; Snell et al., 2022b), and then use the predicted reward to estimate the policy gradient (PG)[1].

$$\nabla_\theta V(\pi_\theta) \approx \frac{1}{n} \sum_{i=1}^n \mathbb{E}_{a \sim \pi_\theta(a|x_i)} [\nabla_\theta \log \pi_\theta(a|x_i)\hat{q}(x_i,a)].$$

Oftentimes, an accurate regression for OPL is difficult to obtain when the relation between prompts and reward is complex. This is because the reward observation is partial and covariate shift arises between the logging policy ($\pi_0$) and the target policy ($\pi_\theta$). If the learned regression model $\hat{q}$ is inaccurate, the estimated PG can be heavily biased (Swaminathan & Joachims, 2015).

**Importance sampling (IS) (Swaminathan & Joachims, 2015).** Instead of using potentially inaccurate regression, IS corrects the distribution shift between $\pi_0$ and $\pi_\theta$ by reweighing the observations:

$$\nabla_\theta V(\pi_\theta) \approx \frac{1}{n} \sum_{i=1}^n \frac{\pi_\theta(a_i|x_i)}{\pi_0(a_i|x_i)} \nabla_\theta \log \pi_\theta(a_i|x_i)r_i.$$

IS is unbiased under the *action support* condition, i.e., $\forall(x,a) \in \mathcal{X} \times \mathcal{A}, \pi_\theta(a|x) > 0 \implies \pi_0(a|x) > 0$. However, IS produces considerable bias due to the violation of the condition (deficient support) (Sachdeva et al., 2020) and extremely high variance due to large importance weight (Saito et al., 2023; 2024; Sachdeva et al., 2024), which are likely when the action space is large. The key shortcoming here is that the typical methods treat each prompt independently and discard the rich information about the generated sentence when estimating the policy gradient.

## 4 PROPOSAL: DIRECT SENTENCE OFF-POLICY GRADIENT (DSO)

The key idea is to make the most of the information about the generated sentence by **taking the policy gradient directly in the sentence space** as follows.

$$\nabla_\theta V(\pi_\theta) = \mathbb{E}_{p(x)\pi_\theta(s|x)}[\nabla_\theta \log \pi_\theta(s|x)q(x,s)].$$

---

[1]The estimation target is the *true* PG defined as $\nabla_\theta V(\pi_\theta) = \mathbb{E}_{p(x)\pi_\theta(a|x)p(r|x,a)}[\nabla_\theta \log \pi_\theta(a|x)r]$.

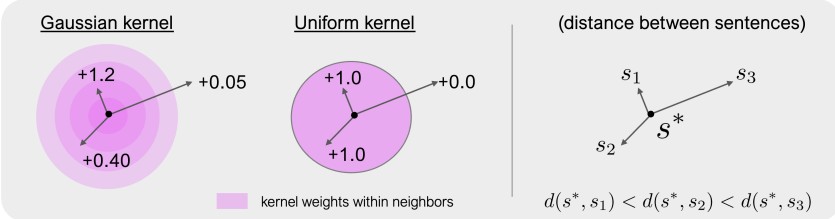

Figure 2: **Examples of the kernel weights and (soft) rejection sampling in the marginalized sentence space.** DSO implicitly augments the data to take the observations for the neighboring sentences into account. (Left) uses a smooth kernel like a Gaussian kernel, and (Right) uses a piecewise constant kernel like a uniform kernel.

Even when we parameterize the policy in the prompt space, this is conceptually possible because we can write the sentence distribution and the score function as $\pi_\theta(s|x) = \sum_{a \in \mathcal{A}} p_{\text{LLM}}(s|x, a)\pi_\theta(a|x)$ and $\nabla_\theta \log \pi_\theta(s|x) = \mathbb{E}_{\pi_\theta(a|x,s)}[\nabla_\theta \log \pi_\theta(a|x)]$, respectively (See Appendix D.1 for the derivation). However, one potential concern of this approach is that we may suffer from data sparsity when estimating the gradient for each sentence $s$, as sentences are high-dimensional. Thus, we further consider taking the gradient in the **marginalized sentence space** to enable data-efficient OPE as

$$\nabla_\theta V(\pi_\theta) = \mathbb{E}_{p(x)\pi_\theta(\phi(s)|x)}[\nabla_\theta \log \pi_\theta(\phi(s)|x)q^{\pi_\theta}(x, \phi(s))],$$

where $\phi(s) \in \Phi(\mathcal{S})$ is the kernel-based neighbors of sentence $s$. Its probability density, policy distribution, and expected reward are defined as follows.

- $\mathbb{P}(\phi(s)|\cdot) := \int_{s' \in \mathcal{S}} K(s', s;\, x, \tau)\mathbb{P}(s'|\cdot)ds',\ \forall \mathbb{P}.$    (marginal density)

- $\pi(\phi(s)|x) := \sum_{a \in \mathcal{A}} p_{\text{LLM}}(\phi(s)|x, a)\pi(a|x),\ \forall \pi.$    (policy marginal distribution)

- $q^\pi(x, \phi(s)) := \int_{s' \in \mathcal{S}} \frac{K(s, s';\, x, \tau)\pi(s'|x)}{\pi(\phi(s)|x)}q(x, s')ds',\ \forall \pi.$    (expected reward)

$K(\cdot)$ is a kernel function, which must satisfy $\int_{s' \in \mathcal{S}} K(s', s; x, \tau) = 1$, and $\tau$ is a bandwidth hyperparameter that controls the magnitude of marginalization. The intuition behind DSO is to implicitly augment the data by taking the observations for the neighboring sentences into account, as illustrated in Figure 2. Specifically, when using a smooth kernel like a Gaussian kernel, neighboring sentences are weighted proportional to $K(s', s;\, x, \tau) \propto \exp(-d(s, s'))$, where $d(s, s')$ is the distance between two sentences (e.g., sentence embedding distance). In contrast, when using a piecewise constant kernel like a uniform kernel, all the sentences within a certain threshold is equally weighted, while all the others are rejected with the weight of 0.

To estimate the policy gradient in the marginalized sentence space induced by kernels, **_Direct Sentence Off-policy Gradient_** (DSO) applies IS as follows.

$$\nabla_\theta V(\pi_\theta) \approx \frac{1}{n} \sum_{i=1}^{n} \underbrace{\frac{\pi_\theta(\phi(s_i)|x_i)}{\pi_0(\phi(s_i)|x_i)}}_{:=w(\phi(s_i), x_i)} \nabla_\theta \log \pi_\theta(\phi(s_i)|x_i)\, r_i.$$

By applying IS on the marginalized sentence space ($\Phi(\mathcal{S})$), DSO avoids large importance weights, making large-scale OPL more scalable regarding the number of candidate prompts, while keeping the bias small by leveraging the similarity among sentences. Moreover, even though we observe only a single prompt in the original logged data, DSO can further distribute the reward observation among multiple prompts that generate similar sentences. This *implicit data augmentation* among multiple counterfactual prompts also contributes to reducing variance. While the precise computation of the marginal importance weight ($w(\phi(s), x)$) and the score function ($\nabla_\theta \pi_\theta(\phi(s)|x)$) seems non-trivial, below we present how to train a model to estimate these distributions in a tractable way.

## 4.1 ESTIMATION OF THE WEIGHTED SCORE FUNCTION

The key trick of DSO is to use the following expression of the weighted score function:

$$w(\phi(s), x)\nabla_\theta \log \pi_\theta(\phi(s)|x) = \mathbb{E}_{(a,s')\sim\pi_\theta(a|x)p_{\text{LLM}}(s'|x,a)}\left[\frac{K(s,s';\ x,\tau)\nabla_\theta \log \pi_\theta(a|x)}{\pi_0(\phi(s)|x)}\right].$$

We provide the derivation in Appendix D.2. This expression indicates that DSO can be seen as performing soft rejection sampling on the data $(a, s')$ augmented by $\pi_\theta$, while correcting the bias in the logged data by applying the inverse propensity of $\pi_0$ in the marginalized sentence space. The above equation also suggests that our estimation problem of the weighted score function is reduced to only the estimation of $\pi_0(\phi(s)|x)$. This is useful, as $\pi_0(\phi(s)|x)$ does not depend on the parameterized policy ($\pi_\theta$), and it thus suffices to fit a marginal density model only once before running the policy gradient method. Because the marginal distribution is defined as $\pi_0(\phi(s)|x) = \mathbb{E}_{\pi_0(s'|x)}[K(s, s';\ x, \tau)]$, we can estimate the marginal density via the monte-carlo sampling as

$$\pi_0(\phi(s_i)|x_i) \approx \frac{1}{m}\sum_{j=1}^m \mathbb{E}_{s_j\sim\pi_0(s_j|x)}[K(s_i, s_j; x_i, \tau)],$$

where $m$ is the number of the monte-carlo samples. Similarly, we can also estimate the marginal density with function approximation ($f_\psi(x, s) \approx \pi_0(\phi(s)|x)$) using the following loss:

$$\ell(f_\psi) \approx \frac{1}{n}\sum_{i=1}^n \mathbb{E}_{(s,s')\sim\pi_0(s|x_i)\pi_0(s'|x_i)}[(f_\psi(x_i, s) - K(s, s';\ x_i, \tau))^2].$$

Since the computation of this loss does not scale with the size of the action space $|\mathcal{A}|$, we can easily apply DSO even when the action (i.e., prompt) space is large.

## 4.2 THEORETICAL ANALYSIS

Here, we analyze the bias and variance of the DSO estimator (the proofs are in Appendix D). We first introduce a new condition about support in the marginalized sentence space.

**Definition 1.** *(Similar sentence support) Similar sentence support is satisfied when $\pi_\theta(\phi(s)|x) > 0 \implies \pi_0(\phi(s)|x) > 0$ holds for all $(x, \phi(s)) \in \mathcal{X} \times \Phi(\mathcal{S})$.*

The similar sentence support condition relaxes the action support condition of IS. That is, because we have $\pi(\phi(s)|x) = \sum_{a\in\mathcal{A}} p_{\text{LLM}}(\phi(s)|a, x)\pi(a|x)$ by definition, the similar sentence support condition is always satisfied when the action support condition is satisfied. This means that deficient support under the similar sentence support is more unlikely happening compared to the action support. Under this condition, we have the following degree of bias.

**Theorem 1.** *(Bias of DSO) When the similar sentence support is satisfied, the bias is*

$$\text{Bias}((\widehat{\nabla_\theta V})_{\text{DSO}}) = \mathbb{E}_{\pi_\theta(\phi(s)|x)}[\nabla_\theta \log \pi_\theta(\phi(s)|x)\Delta_q(\pi_\theta, \pi_0;\ x, \phi(s))]$$
$$+ \mathbb{E}_{\pi_0(\phi(s)|x)\pi_0(s'|x,\phi(s))}[\Delta_{(w\nabla_\theta)}(\phi(s'), \phi(s); x)q(x, s')]$$
$$+ \mathbb{E}_{\pi_\theta(\phi(s)|x)\pi_\theta(s'|x,\phi(s))}[\Delta_{(\nabla_\theta)}(\phi(s), s'; x)q(x, s')].$$

*where $\Delta_q(\pi_\theta, \pi_0;\ x, \phi(s))$ is the difference of $\hat{q}^\pi(x, \phi(s))$ between $\pi_\theta$ and $\pi_0$. $\Delta_{(w\nabla_\theta)}(\phi(s'), \phi(s); x)$ is the difference of weighted score function between $\phi(s')$ and $\phi(s)$. $\Delta_{(\nabla_\theta)}(\phi(s), s'; x)$ is the difference between the score function of $\phi(s)$ and $s'$, which is equivalent to the difference of $\mathbb{E}_{\pi_\theta(a|x,\phi(s))}[\nabla_\theta \log \pi_\theta(a|x)]$ and $\mathbb{E}_{\pi_\theta(a|x,s')}[\nabla_\theta \log \pi_\theta(a|x)]$.*

Theorem 1 suggests that the bias of DSO comes from three factors. The first term is the dominant term, which arises from the *within-neighbor* reward shift (i.e., the difference between $q^{\pi_0}(x, \phi(s))$ and $q^{\pi_\theta}(x, \phi(s))$), as illustrated in Figure 3. This term becomes small in two cases: (i) when reward does not change too much within $\phi(s)$, and (ii) when the within-cluster distribution shift of $\pi(s'|x, \phi(s))$ is small. Either case is satisfied when the radius of neighbors (i.e., kernel bandwidth hyperparameter $\tau$) is small. At the same time, smooth kernels like a Gaussian kernel are also useful, as they allocates

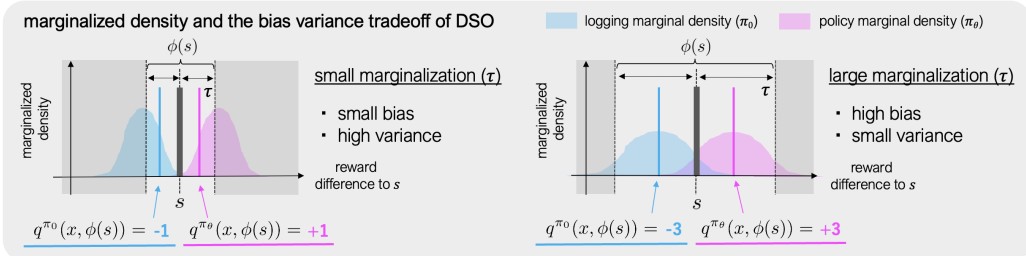

Figure 3: **Bias-variance tradeoff of DSO and its relations to the bandwidth hyperparameter ($\tau$) of a kernel function**: When $\tau$ is large, the overlap between the logging policy ($\pi_0$) and the current policy ($\pi_\theta$) within $\phi(s)$ becomes large, thus the scale of the importance weight becomes small. This contributes to reducing the variance compared to naive IS. In contrast, a small value of $\tau$ helps keep the bias small, as the within-neighbor reward shift (i.e., the difference between $q^{\pi_0}(x, \phi(s))$ and $q^{\pi_\theta}(x, \phi(s))$) becomes small. The gray regions are rejected when using a uniform kernel.

larger weights to similar sentences depending on the distance from the pivotal sentence. In contrast, the second and third terms are caused by calculating the gradient in the marginalized sentence space ($\Phi(\mathcal{S})$) instead of the original sentence space ($\mathcal{S}$). These terms also become small when the bandwidth hyperparameter $\tau$ is small. Thus, a small value of $\tau$ is preferable in reducing the bias.

Next, we have the following degree of variance using DSO.

> **Theorem 2.** *(Variance of DSO) When the similar sentence support is satisfied, the conditional variance is expressed as*
>
> $$n\mathbb{V}_{\mathcal{D}|x}((\widehat{\nabla_\theta V})_{\text{DSO}}) = \mathbb{V}_{\pi_0(s|x)}(w(\phi(s), x)\nabla_\theta \log \pi_\theta(\phi(s)|x)q(x, s))$$
> $$+ \mathbb{E}_{p(x)\pi_0(s|x)}[(w(\phi(s), x))^2(\nabla_\theta \log \pi_\theta(\phi(s)|x))^2\sigma^2(x, s)].$$
>
> *Compared to the naive (action) IS, the importance weight and the gradient reduce the variance by $\mathbb{E}_{\pi_0(\phi(s)|x)}[\mathbb{V}_{\pi_0(a|x,\phi(s))}(w(a, x))]$ and $\mathbb{E}_{\pi_0(\phi(s)|x)}[\mathbb{V}_{\pi_0(a|x,\phi(s))}(\nabla_\theta \log \pi_\theta(a|x))]$, respectively, where $w(x, a)$ is the action importance weight.*

Theorem 2 suggests that DSO gains variance reduction from two sources: $\nabla_\theta \log \pi_\theta(\phi(s)|x)$ and $w(\phi(s)|x)$. The first variance reduction of $\nabla_\theta \log \pi_\theta(\phi(s)|x)$ comes from the fact that the sentence-based score function is expressed as $\mathbb{E}_{\pi_\theta(a|x,\phi(s))}[\nabla_\theta \log \pi_\theta(a|x)]$, demonstrating the benefit of applying the implicit data augmentation and soft rejection sampling (instead of applying hard rejection sampling). The variance reduction becomes especially large when multiple different prompts result in similar sentences; thus, $\pi_\theta(a|x, \phi(s))$ becomes adequately stochastic. Moreover, by using $w(\phi(s), x)$ instead of $w(a, x)$, we can expect a significant variance reduction as we avoid the variance caused by the within-neighbor importance weight, i.e., $w(a, x; \phi(s)) := \pi_\theta(a|x, \phi(s))/\pi_0(a|x, \phi(s))$. This means that a larger value of $\tau$ (i.e., the radius of neighbors) leads to a larger variance reduction. Together with the analysis of bias, we can see that the value of $\tau$ plays an important role in trading off the bias and variance of DSO, as shown in Figure 3. Later in the experiment section, we study how the performance changes with varying values of the bandwidth hyperparameter $\tau$.

## 5 BENCHMARKS AND OPEN-SOURCE SOFTWARE

Due to the lack of existing benchmark suites for OPL of prompt policies, we implemented and will release open-source software called **OfflinePrompts**. This benchmark suite come with two settings: *synthetic* and *full-LLM* to enable extensive and reproducible experiments. In particular, the full-LLM benchmark simulates movie recommendation tasks with personalized sentence descriptions based on the (sentence-augmented) MovieLens dataset (Harper & Konstan, 2015) (See Appendix B for the details). Moreover, OfflinePrompts also enables prompt tuning on users' own logged data, facilitating the practical application of OPL. Appendix A summarizes related benchmarks and the distinctive features of our software. Appendix F also demonstrates the easy-to-use APIs of OfflinePrompts.

# 6 SYNTHETIC EXPERIMENTS

We first evaluate the proposed DSO approach on synthetic benchmarks in OfflinePrompts, since they allow us to explore a wide range of conditions.[2]

## 6.1 EXPERIMENT SETTING

To generate candidate actions, we first sample 5-dimensional embedding $e_a$, from a normal distribution. Each embedding $e_a$ is a deterministic embedding associated with an action $a$. Then, to generate logged data, we sample 5-dimensional user and query vectors from a multivariate normal distribution. Next, for each query-action pair $(q, a)$, we sample 5-dimensional sentence embeddings $s$ as

$$s \sim \mathcal{N}(f_s(q, e_a), \sigma_s^2), \quad f_s(q, e_a) = c \cdot \text{sine}(q^\top M_q + e_a^\top M_e),$$

where $M_q$ and $M_e$ are coefficient matrices sampled from a uniform distribution. $c = 5.0$ is a scaling factor and $\sigma_s = 1.0$ is the noise level of the action-output mapping. By using the sine function, we simulate a situation where two different prompts ($e_a$) can result in a similar sentence ($s$), while preserving the smoothness between the prompt and sentence embedding spaces. Then, a user responds to the generated sentence ($s$) with the following reward function:

$$r \sim \mathcal{N}(f_r(x, s), \sigma_r^2), \quad f_r(x, s) = (u^\top M_u + q^\top M_q)M_s s^\top,$$

where $M_u$, $M_q$, and $M_s$ are the coefficient matrices and $\sigma_r$ is the reward noise.

We generate logged data with the following softmax logging policy: $\pi_0(a|x) := \exp(\beta_0 \hat{R}_0(x, a))/(\sum_{a \in \mathcal{A}} \exp(\beta_0 \hat{R}_0(x, a)))$. $\hat{R}_0$ is the base reward model, trained on $n_0 = 10000$ of data points collected by the uniform random policy. $\beta_0 = 1.0$ is the inverse temperature.

We compare **DSO** to four baselines: **regression, IS, DR, and POTEC**. DR (Dudík et al., 2011) combines IS and regression efficiently. POTEC (Saito et al., 2024) employs a two-stage policy learning, which first chooses which cluster to use via DR and then chooses which action within the cluster to use via regression. All the baselines estimate the gradient in the action space. For the metrics to compare the OPL methods, we use the *optimality* of the learned policy, defined as $(V(\pi) - V(\pi_{\text{unif}}))/V(\pi_{\text{opt}} - V(\pi_{\text{unif}}))$, where $\pi_{\text{opt}}$ is the optimal policy and $\pi_{\text{unif}}$ is the uniform random policy. The definitions of DR and POTEC, and the implementation details are in Appendix C.

The experiment varies the following configurations (the **bold** font represents the default value): (1) **data size**: $n \in \{500, 1000, 2000, 4000, \mathbf{\underline{8000}}\}$, (2) **number of candidate actions**: $|\mathcal{A}| \in \{10, 50, 100, 500, \mathbf{\underline{1000}}\}$, and (3) **reward noises**: $\sigma_r \in \{0.0, \mathbf{\underline{1.0}}, 2.0, 3.0\}$. For the ablation of DSO, we additionally report the results with the varying **bandwidth hyperparameters** of $\tau \in \{0.5, \mathbf{\underline{1.0}}, 2.0, 4.0\}$, $\{\mathbf{\underline{w/}}$ and w/o$\}$ **function approximation** of the marginal density, and two different kernels, $\{\mathbf{\underline{Gaussian}}$ and uniform$\}$. When not using function approximation, we estimate the marginal density via monte-carlo sampling with $m = 100$ samples. Finally, to evaluate the robustness of DSO to the accuracy of the distance measure in the kernel, we add noise sampled from a normal distribution with std $\Delta_s = 1.0$ to the sentence embeddings. We report the mean and standard deviation of the performance based on the results with 20 random seeds.

## 6.2 RESULT

Figure 4 compares the policy learning results of the OPL methods with varying data sizes ($n$), number of candidate actions ($|\mathcal{A}|$), and reward noises ($\sigma_r$), respectively. The results demonstrate that DSO works particularly well in challenging scenarios where the baselines fall short due to variance. Specifically, while we observe a sharp drop of performance for the baselines when the action space is large ($|\mathcal{A}| \geq 500$) and reward noise is large ($\sigma_r \geq 1.0$), DSO maintains a favorable performance even under these configurations. Moreover, comparing the performance with $|\mathcal{A}| = 1000$ and $\sigma_r = 1.0$, we observe that the performance of DSO at $n = 500$ outperforms that of the baselines at $n = 8000$. This indicates that DSO is far more data-efficient than the baselines when the action space is large, leveraging the similarity among sentences via kernels and performing implicit data augmentation.

Next, we study how the choice of kernels affects the performance of DSO, as shown in Figure 5. The results tell us several interesting findings: using (1) a Gaussian kernel and (2) function approximation

---

[2]Our code will be available at a GitHub repository upon publication.

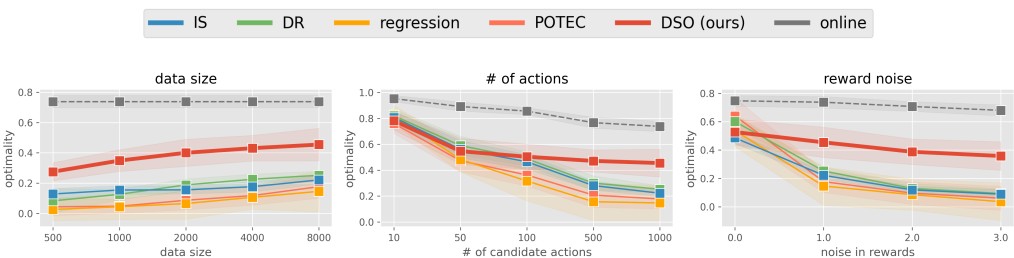

Figure 4: Comparing the performance of the policies learned by various OPL methods with (Left) **varying data sizes** ($n$), (Middle) **varying number of candidate actions** ($|\mathcal{A}|$), and (Right) **varying reward noises** ($\sigma_r$). DSO uses a Gaussian kernel and function approximation of $\pi_0(\phi(s)|x)$.

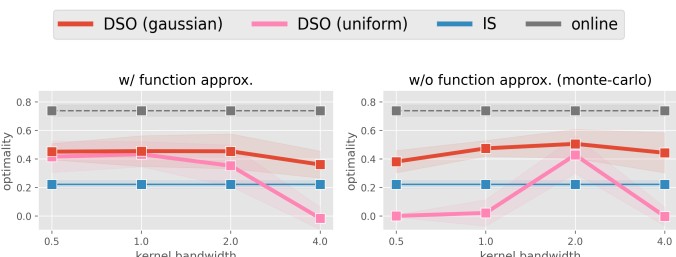

Figure 5: Ablation results of DSO with **varying bandwidth hyperparameters** ($\tau$), **w/ and w/o function approximation of** $\pi_0(\phi(s)|x)$, and **two kernels, Gaussian and uniform**.

improve the robustness of DSO to the choice of bandwidth hyperparameter $\tau$. The first observation is evident from the fact that a Gaussian kernel allocates larger weights to closer sentences compared to a uniform kernel. However, when using monte-carlo estimation, we observe that even a Gaussian kernel needs careful tuning of $\tau$, where a small value of $\tau$ incurs high variance and a large value of $\tau$ produces non-negligible bias. In contrast, by using function approximation, we can avoid a small value of $\hat{\pi}_0(\phi(s)|x)$, which contributes to the variance reduction[3]. Therefore, using function approximation helps improve the robustness to a small value of $\tau$, and we do not need extensive hyperparameter tuning of $\tau$. This implies that DSO is applicable to practical situations, where a pre-trained model of $\hat{\pi}_0(\phi(s)|x)$ can provide substantial efficiency gains.

# 7 FULL-LLM EXPERIMENT WITH MOVIELENS

This section compares OPL methods in *a personalized generation task of movie descriptions* using the MovieLens-10M (Harper & Konstan, 2015) dataset. The MovieLens dataset contains 10M ratings between 71,567 users and 10,681 movies. To use this data in our personalized sentence generation task, we first augment the data by generating a (general) movie description using Mistral-7B (Jiang et al., 2023). Then, we train a sentence-based reward simulator on the augmented dataset using DistilBert (Sanh et al., 2019). After obtaining a reward simulator, we collect the logged data in the following procedure. First, we randomly sample a user ($u$) and a movie (query) ($q$) as a context ($x$). Next, a logging policy ($\pi_0$) chooses which prompt ($a$) to use in the sentence generation task. Then, a frozen LLM generates sentence $s$, taking the prompt $a$ and query $q$ as the input. Finally, we generate a reward ($r$) using the reward simulator. Appendix B.2 and Figure 10 describe the workflow of OPL and that of pre-training a reward simulator in detail.

In the full-LLM experiment, we define the logging policy by applying the softmax function on top of the logits learned by the online policy, where we set the inverse temperature hyperparameter to be

---

[3]This is because, for example, when the true marginal density is 1e-5, estimating it as 1e-5 and 1e-4 does not change the MSE loss too much. However, in terms of variance, 1e-4 and 1e-5 make a significant difference. Using function approximation, we can avoid being too precise about small values of the marginal density.

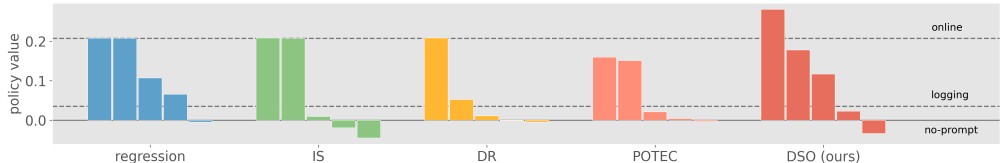

Figure 6: **Performance comparison of OPL methods in the full-LLM experiment.** The policy value indicates how much improvement of reward we have by using a (learned) prompt policy compared to the sentence generation without prompts (called *no-prompt* baseline). From the top, the horizontal lines refer to the value of the online policy, logging policy, and no-prompt baseline. The results are based on 5 random seeds and are ordered by the performances.

$\beta_0 = 0.2$. The candidate prompts ($\mathcal{A}$) are retrieved from relatedwords.io with keywords {"movie", "genre", "culture"}, where $|\mathcal{A}| = 1000$. The reward is defined as $10 \times (q(x, s(a)) - q(x, s(\emptyset)))$, where $q(\cdot)$ is the [0, 1]-score simulated by the aforementioned DistilBert sentence discriminator and $s(\emptyset)$ is the sentence generated without adding prompts. The data size is $n = 50000$. For DSO, we use a Gaussian kernel with $\tau = 1.0$ to estimate the logging marginal density. The distance between two sentences are measured by the sentence embeddings obtained from the *frozen* Mistral-7B model (see Appendix C for the details). We report the results with 5 random seeds.

**Result** Figure 6 compares the performance of the OPL methods by the degree of improvement that the learned policy observed over the sentences generated without prompts (which we call *no-prompt* baseline).[4] The results indicate that DSO often improves the effectiveness of the sentences more than other IS-involved OPL methods, by effectively leveraging the information about similar sentences. Specifically, DSO is more resilient to performance corruption than IS by substantially reducing the variance. It should also be worth noting that this result is observed for the off-the-shelf embeddings of sentences, which do not require extensive tuning of the embedding model. This minimizes the difficulty in applying the proposed OPL method in practice. However, learning (application-specific) embeddings that further improve the performance of DSO is an interesting direction for future work.

## 8 CONCLUSION AND FUTURE WORK

This paper studied how to use naturally logged user feedback to optimize a prompt policy for language generation. We started by formulating the problem as OPL of contextual bandits with auxiliary outputs. Then, we pointed out the limitations of the naive approaches – (1) existing OPL methods often suffer from the large action space of prompts and (2) even though we observe generated sentences, existing methods do not use the information about these sentences. To overcome these shortfalls, we proposed *Direct Setence Off-policy gradient* (DSO), which applies importance sampling taking the similarity of sentences into account. We also show the effectiveness of the proposed approach in both theoretical and empirical ways. Furthermore, our benchmarks suite called OfflinePrompts, provided as open-source software, accelerates future research and practical application of prompt-guided language generation from logged bandit feedback.

As a remark, deriving a DR-style variant, which introduces a control variate for further variance reduction (Dudík et al., 2011), is non-trivial for DSO, as we discuss in detail in Appendix E. Studying a way to efficiently combine IS and regression in the DSO framework would be a promising future direction. Additionally, a good representation or distance measure of sentences that further improves the performance of DSO would also be worth exploring. Finally, applying a similar idea to other generative AI applications, such as text-to-image diffusion models (Saharia et al., 2022), and extending the benchmark by publishing relevant real-world data can be an interesting future work.

---

[4]While we observe some instability in the policy value for all the compared methods, this simply implies that the task of finding an effective prompt from the large candidate sets of prompts is challenging. This is because most prompts do not make so much difference to the no-prompt baseline, while only a few prompts can be effective, as shown in Figure 11 in the appendix. A similar instability issue is also observed for the online policy.

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

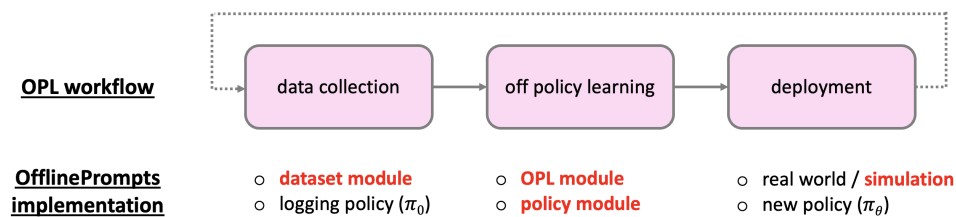

Figure 7: **Off-policy learning (OPL) workflow and OfflinePrompts modules**.

## A  EXTENDED RELATED WORK

**(Online) linear and kernelized contextual bandits.**    Linear bandits (Li et al., 2010; Agrawal & Goyal, 2013; Rusmevichientong & Tsitsiklis, 2010) and kernelized contextual bandits (Chowdhury & Gopalan, 2017; Valko et al., 2013; Zhou et al., 2020) is relevant to ours in using the similarity among actions and rewards for improving data efficiency. Specifically, linear bandits assume that the reward function is expressed as an inner-product between (non-linear representations of) context features and action-specific coefficients and aims to learn the linear action features (Li et al., 2010). By assuming the linear structure in the reward function, the corresponding bandit algorithm makes the exploration more efficient than treating each action independently. Similarly, kernelized bandits (Valko et al., 2013) generalize the idea of leveraging similarity among action representations under less restrictive assumptions on the rewards. Specifically, it assumes that similar features can result in similar rewards without assuming a linear structure and implicitly augments the reward observation using the Reproducing kernel Hilbert space (RKHS). Our paper demonstrates that leveraging the similarity among auxiliary outputs (i.e., sentences) of action can improve the data efficiency in the *offline* learning setting, not only limited to the *online exploration* discussed in the existing literature.

**Offline reinforcement learning (Offline RL) for dialog generation.**    Offline RL (Levine et al., 2020) has emerged as a new paradigm for fine-tuning language models in dialog systems (Jaques et al., 2020; Snell et al., 2022a;b; Verma et al., 2022). Among them, Snell et al. (2022a) and Verma et al. (2022) focus on *goal-oriented* dialog system, which aims to solve some specific tasks by combining RL-based planning and dialog generation. Jaques et al. (2020) and Snell et al. (2022a) aim to improve the quality of conversations by maximizing the users' sentiment signals observed in text or interface (e.g., thumb up). While these works are relevant to ours in using offline data, our work has several distinctions over existing works. First, while existing work focuses on RL-based fine-tuning, which requires expensive computation and is affordable only for the companies releasing pre-trained models, our work considers prompt tuning. Since prompt tuning is available for some third-party companies (e.g., advertising agencies) or even individual users that access models through APIs (e.g., ChatGPT), a more diverse population can customize language generation with our framework. Moreover, while existing work formulates the problem as an RL problem and considers only the regression-based approach for policy learning, ours formulates the problem as contextual bandits and considers applying IS in the marginalized sentence space. This makes the bias-variance tradeoff of policy learning more controllable than existing works. Thus, we can expect an improved policy performance, as we have shown in the experiments.

**Relevant open-source softwares and benchmarks.**    There are several open-source libraries for language generation relevant to ours. First, RL4LMs (Ramamurthy et al., 2022) provides a framework for RL-based fine-tuning of LLMs for optimizing language generation for reward maximization. In prompt tuning, OpenPrompt (Ding et al., 2022) works as a testbed for comparing (online) gradient-based prompting strategies with various frozen LLMs. OpenICL (Wu et al., 2023) also benchmarks (more sophisticated) prompt conditioning strategies called *in-context learning* (ICL), such as *chain-of-thoughts* reasoning for solving complex mathematical problems (Wei et al., 2022). Similarly, OpenAgents (Xie et al., 2023) provides an interface for generating text for various real-world web applications using frozen LLMs, especially for the purpose of providing a platform for online RL-based prompt tuning. However, these platforms are not capable of handling logged bandit feedback, and ours are the first to streamline OPL procedures for prompt tuning with naturally collected user feedback data.

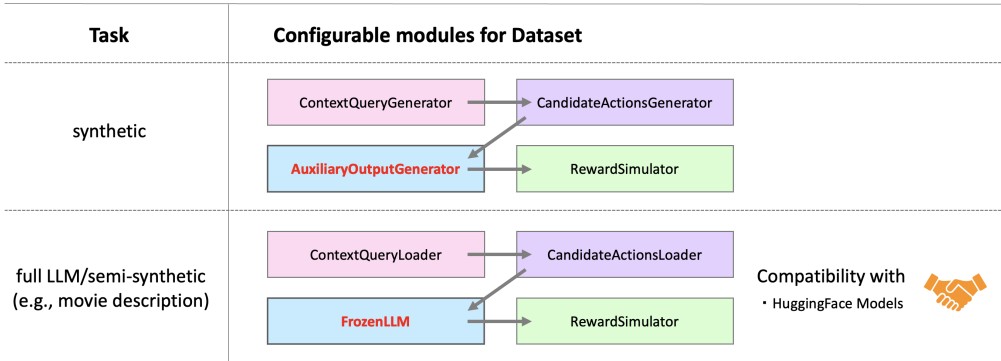

Figure 8: **Two benchmarks (synthetic and full-LLM) of OfflinePrompts with four configurable submodules**. Compared to the existing OPE/OPL frameworks (Saito et al., 2021; Kiyohara et al., 2023a;b), our benchmark is distinctive in providing `AuxiliaryOutputGenerator`/`FrozenLLM` to simulate language generation tasks as contextual bandits with auxiliary outputs.

In an independent field of benchmark study, OpenBanditPipeline (Saito et al., 2021) and SCOPE-RL (Kiyohara et al., 2023a;b) are representative open-source libraries to handle OPE and OPL procedures in contextual bandits and RL. Although these libraries streamline the workflow of using logged data in organized ways, they are not applicable to language generation. Thus, we release a new benchmark suite for OPL of prompt tuning for language generation, putting emphasis on connecting OPL modules and language generation modules, while following the basic design principles of OpenBanditPipeline (Saito et al., 2021) and SCOPE-RL (Kiyohara et al., 2023a;b).

Finally, there is also a benchmark called BanditBench (Nie et al., 2024) , which simulates the LLM-based item recommendations based on the MovieLens (Harper & Konstan, 2015) dataset. While this benchmark uses the same MovieLens dataset for semi-synthetic simulation, the tasks are different from each other. Specifically, BanditBench (Nie et al., 2024) aims to use LLMs as recommender policies that choose items (Yang et al., 2023; Gao et al., 2023), while our work focuses on steering the generation of sentence description with prompts given (already chosen) items as a query.

# B    OFFLINEPROMPTS: OPEN-SOURCE SOFTWARE OF OPL FOR LANGUAGE GENERATION

## B.1    OVERVIEW AND WORKFLOW

The primal goals of *OfflinePrompts* are to (1) provide a standardized benchmark to compare OPL methods and (2) facilitate the smooth implementation of the OPL workflow. For these purposes, OfflinePrompts (1) provides two standardized benchmarks and (2) streamlines the implementation with three modules: dataset, OPL, and policy, as shown in Figure 7. All implementations are based on PyTorch (Paszke et al., 2019). We elaborate on the details of each feature below.

**Dataset module and benchmarks**    OfflinePrompts provides two benchmarks including *synthetic* and *movie description*. First, the synthetic benchmark simulates (general) contextual bandits with auxiliary outputs with feature vectors, without involving language generation tasks. In contrast, the movie description task is a full-LLM, semi-synthetic benchmark, which simulates personalized generation of movie description (i.e., actual language generation) based on the MovieLens datasets (Harper & Konstan, 2015). These benchmarks can be used for separate purposes. The synthetic benchmark is lighter and more suitable for extensive studies of how the performance of OPL methods changes with various configurations than the actual full-LLM benchmark. In contrast, the movie description benchmark is preferable to see the performance of OPL methods in more realistic settings than the synthetic benchmark. The key remark is that the movie description benchmark is the first benchmark for prompt-guided language generation from logged user feedback.

Both benchmarks provide a standardized setting and configurable submodules to control the data generation process. Specifically, as illustrated in Figure 8, each benchmark consists of four submodules: `ContextQueryModule`, `CandidateActionsModule`, `AuxiliaryOutputGenerator`/`FrozenLLM`, and `RewardSimulator`. Compared to the existing OPE/OPL benchmarks (Saito et al., 2021; Kiyohara et al., 2023a;b), our benchmark is distinctive in modeling `AuxiliaryOutputGenerator`/`FrozenLLM`, which enable us to simulate language generation tasks as contextual bandits with auxiliary outputs. Moreover, since our `FrozenLLM` and `RewardSimulator` modules are compatible with HuggingFace (Wolf et al., 2019), users can easily employ various language models in the full-LLM experiments. The semi-synthetic/full-LLM dataset module can also load custom dataset in a manner similar to the movie description benchmark. We believe this feature of OfflinePrompts also facilitates practical applications of prompt tuning from naturally logged feedback.

**OPL and policy modules** Figure 9 summarizes the implementation choices of OPL modules:

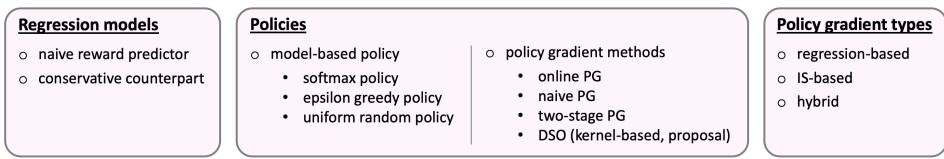

Figure 9: **Implementation choices of OPL modules of OfflinePrompts**.

As shown above, we implement two regression models (naive and conservative), three model-based policies (softmax, epsilon-greedy, and uniform random), and four policy gradient methods (online, naive, two-stage, DSO), and three gradient types (regression-based, IS-based, and hybrid). Each component is independently configurable. Thus, we can easily try any combination of the above policies and OPL methods. Moreover, by following the abstract base implementation provided in OfflinePrompts, researchers can test their own policies and policy gradient methods.

Example codes for streamlining OPL workflow and customizing each module are available in Appendix F. The documentation of OfflinePrompts, which describes further details of the software, is also available at: (double blind review).

### B.2 TASK DESCRIPTION AND REWARD SIMULATION FOR THE MOVIE DESCRIPTION TASK

We build a semi-synthetic simulator using the MovieLens dataset (Harper & Konstan, 2015) for the personalized generation task of movie descriptions. The movies consist of (partially observed) 5-star ratings between users and items and have movie title information as the metadata. To learn a reward simulator, which generates reward depending on the generated movie description, we first augmented the movielens dataset with item description using a frozen LLM as follows.

1. For each movie, retrieve its title.
2. Then, using zero-shot inference of a frozen LLM, we generate the movie description by providing instruction: `"Broadly describe in a sentence the genres of the movie without including the name or any specifics of the movie. Title: {title of the movie}, Movie description: "`.

In our standardized benchmark, we use Mistral ("mistralai/Mistral-7B-Instruct-v0.2") (Jiang et al., 2023) as the frozen LLM. Once augmenting the dataset with item description, we train a sentence encoder-based collaborative filtering model (as shown in Figure 10 (Top)) in the following procedures.

1. Using movielens dataset *without* item description, we first train a naive neural collaborative filtering (CF) model (He et al., 2017), which uses user and item id embeddings.
2. Initialize the encoder-based CF model, which uses user id embeddings and encoded item description features, with the user id embeddings learned by naive CF model.
3. Finetune the encoder-based CF model with the (augmented) movielens dataset *with* item description.

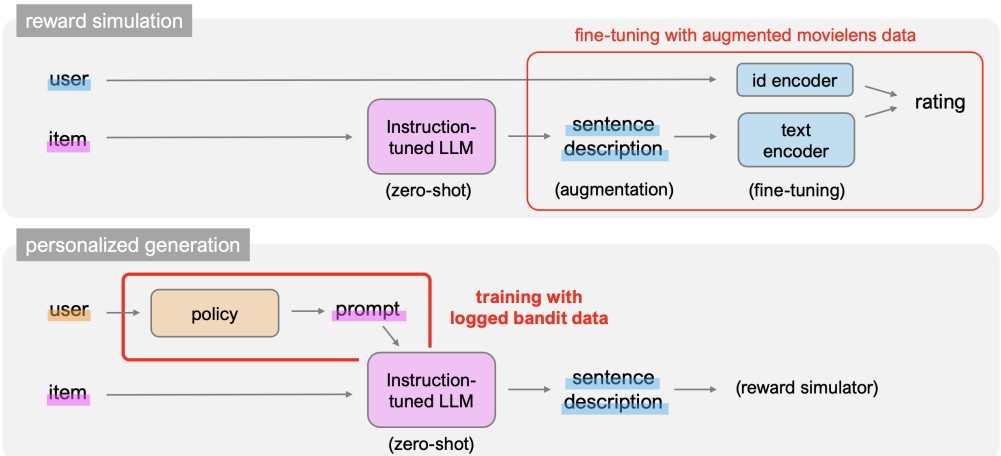

Figure 10: **Procedures of reward simulation (Top) and personalized sentence generation (Bottom)**. The reward simulator uses a sentence encoder to get item embeddings. In the personalized sentence generation task, a policy aims to identify a suitable prompt (e.g., genres of the movie) for each user so the generated sentence aligns with the user-dependent preference.

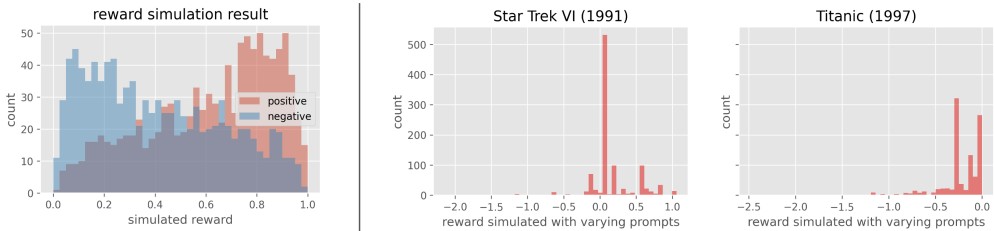

Figure 11: **The reward simulation results on the MovieLens dataset (Harper & Konstan, 2015)**. (Left) Showing the *original* reward simulated by the fine-tuned DistilBert model (Sanh et al., 2019) for 2000 samples of the validation data. "positive" indicates the data that originally received the rating of 5 by users, and "negative" indicates the data that received 0-3 ratings. (Right) Showing the *normalized* reward generated for a single user and two movies with varying prompts. This demonstrates how much reward difference each prompt can make compared to without prompts (which we call *prompt effect*), suggesting that effective prompts are sparse among the candidate set, and we have skewed distribution on the prompt effect.

The default reward simulator in our benchmark uses the fine-tuned DistilBert model (Sanh et al., 2019) as the item description encoder, and the user and item embeddings are set to be 20 dimensional. Note that, before training the models, we preprocess the MovieLens-10M to have binary labels – the rating of 5 is positive, and the ratings of 0-3 are negative. Then, we prune the dataset so that the dataset has balanced positive and negative labels, and each user and item has at least 10 positive and negative labels. After processing the data, 36,395 users, 4,796 items, and 2,316,912 ratings remained. When using the fine-tuned model as the reward simulator in our benchmark, we use the following *normalized* reward: $10 \times (q(x, s(a)) - q(x, s(\emptyset)))$, where $q(\cdot)$ is the original $[0, 1]$-score simulated by the model. $s(a)$ is the sentence generated by the prompt $a$, and $s(\emptyset)$ is the sentence generated without any prompt. We report the reward simulation results in Figure 11.

Finally, we simulate the data generation of the movie description task as follows.

1. Randomly sample user and item id and let the user embedding learned by the naive CF as the user context $u$. We also let the title of the movie be query $q$. This is handled by `ContextQueryLoader`. ($x$)

2. (A policy chooses which prompt to use, taking the user and query embeddings as inputs.) ($a$)

Figure 12: **Example of sentences generated with varying prompts and their reward simulated in the full-LLM benchmark**. We use the frozen Mistral-7B (Jiang et al., 2023) model to generate descriptions of the movie Star Trek VI (1991) with three different prompts, { movie, scifi, tragedy } and highlight sentences that differ from the baseline generated without prompts. The red font indicates the reward simulated by the DistilBert model fine-tuned on the MovieLens dataset. While abstract keywords like "movie" do not make much difference, more specific keywords like "scifi" or "tragedy" can be impactful in the reward simulation.

3. `FrozenLLM` takes query and prompt as input in the following instruction: `"Broadly describe in a sentence the genres of the movie without including the name or any specifics of the movie. Title: {title of the movie}, Keyword: {prompt} Movie description: "` and generate movie description. ($s$).

4. `RewardSimulator` simulates reward by taking user id, item id, and the generated sentence as inputs. User and item ids are used to retrieve user-and item-specific bias terms. ($r$)

Note that by pre-training the reward simulator with item descriptions, we expect the model to learn matchings between user preferences and movie genres (e.g., user A prefers sci-fi movies). We expect this differentiates the reward among varying prompts in the movie description task – e.g., for sci-fi lovers, we should focus on the sci-fi aspects rather than the romance aspects of a movie. The goal of OPL task is to identify specific features or keywords that generate suitable sentence for each user from the logged data. For reference, Figure 12 shows the example of sentences generated with varying prompts and their rewards simulated in our benchmark.

### B.3 DATA GENERATION PROCESS FOR THE SYNTHETIC BENCHMARK

The synthetic benchmark simulates the contextual bandits with auxiliary output using feature vectors without involving actual language generation. Specifically, the synthetic benchmark generates the logged data in the following process:

1. Sample size $n$ of context and query from `ContextQueryGenerator`. ($x$)

2. Sample embeddings ($e_a$) for size $|\mathcal{A}|$ of actions to define a candidate set of actions. Then, for each context, sample action from the candidates with some logging policy. ($a$)

3. Input both query and the chosen action to `AuxiliaryOutputGenerator` to generate a feature vector as an auxiliary output. The auxiliary output corresponds to output sentence in language generation tasks. ($s$)

4. Finally, simulate base reward $R(x, s)$ by inputting context and auxiliary output into `RewardSimulator`. Then, sample reward from a normal distribution $\mathcal{N}(R(x, s), \sigma_r)$ for the auxiliary output observed by the logging policy. ($r$)

By running a synthetic experiment, we can easily control and study the effect of various relationship between prompt and sentence (action $a, e$ and auxiliary output $s$) and that between sentence and reward (auxiliary output $s$ and reward $r$) through varying `AuxiliaryOutputGenerator` and `RewardSimulator`, respectively. Therefore, we expect our synthetic benchmark to be a easy-to-use testbed for checking the behaviors of OPL methods before working on a more complex, actual language generation task.

## C  IMPLEMENTATION DETAILS OF EXPERIMENTS

Basically, we follow the default implementation of OfflinePrompts.

### C.1  SYNTHETIC EXPERIMENTS

The policy is parameterized by a two-layer neural network, where the hidden dimension is 100, the activation function is ReLU, and the optimizer is Adam (Kingma, 2014). All single-stage policies (which are used in regression-based, IS-based, and DSO) take $(x, e_a)$ as inputs and generate logit values. The probability of each prompt chosen by the policy is calculated by taking the softmax of the logit values. Similarly, the first stage policy of POTEC takes $(x, e_c)$ as inputs, where $e_c$ is the cluster centers of prompts in the action embedding space. For all IS and DR-type methods, we apply the weight clipping with a maximum of 200. To avoid the extensive tuning of learning rates, we use the one that worked well for the online policy gradient in all the compared methods, which is 5e-4. The regression model ($\hat{q}(x, e_a)$), used for regression-based, DR, and POTEC, and the logging marginal density model used for DSO are parameterized by a two-layer neural network with a 100 dimensional hidden state. The regression model is trained on the logged data with the following MSE loss: $\sum_{i=1}^{n}(r_i - \hat{q}(x_i, a_i))^2$, while the marginal density model is trained by the loss function described in Section 4.1. The learning rates of the regression and the marginal density models are both based on the validation loss, and are set to 1e-4. Note that because $\phi(s)$ ranges within $[-3\tau, 3\tau]$ with probability more than 99% under the Gaussian kernel, we let $\phi(s)$ of the uniform kernel to range $[-3\tau, 3\tau]$ to the corresponding value of $\tau$. Finally, the action clustering used by POTEC is based on $k$-means clustering with $k = 10$, implemented in scikit-learn (Pedregosa et al., 2011).

### C.2  FULL-LLM EXPERIMENT

The implementation of the full-LLM experiment is almost the same as the synthetic experiment. The only difference is that, because $(q, a, s)$ are words or sentences, we applied some encoding to get vectorial embeddings of these variables. We learn the embeddings by the following steps. We first randomly sample 1000 movies and sentences from the (augmented) MovieLens dataset and sample 1000 prompts from the action set. Then, we get the last hidden states of Mistral-7B (Jiang et al., 2023) by providing the following instructions:

- `"Broadly describe in a sentence the genres of the movie without including the name or any specifics of the movie. Title: { title of the movie }"` for each movie ($q$),
- `"Associate the word – { prompt } – in the context of movie genres"` for each prompt ($a$),
- `" { sentence } "` for each sentence $s$.

After obtaining (high-dimensional) embeddings from Mistral-7B, we fit PCA (Maćkiewicz & Ratajczak, 1993) to reduce the dimension of embeddings to 20. In contrast, for the user context $x$, we use 20-dimensional embeddings of user ids learned by (naive) collaborative filtering, which is different from that used in the reward simulator. We use the learning rate of 5e-4 with Adam for policy gradients. The learning rate of the regression and the marginal density models are 1e-4 with Adam.

It is worth mentioning that because the full-LLM benchmark is challenging due to the sparsity of effective prompts as demonstrated in Figure 11, a similar learning instability to the OPL results is

also observed for the online policy (e.g., even the online policy sometimes fall short with a near-zero policy value like 0.01, regardless the choice of the learning rates). Therefore, in the experiment, we picked an online policy that performed well when defining a logging policy, so that meaningful reward signals should be included in the logged data.

### C.3 DOUBLY ROBUST (DR) ESTIMATORS

Here, we provide the details of the DR estimators used in the experiments.

**Doubly Robust (DR) (Dudík et al., 2011).** DR is a hybrid approach, which effectively combines the regression and IS to exploit the benefits of the two.

$$\nabla_\theta V(\pi_\theta) \approx \frac{1}{n} \sum_{i=1}^n \frac{\pi_\theta(a_i|x_i)}{\pi_0(a_i|x_i)} \nabla_\theta \log \pi_\theta(a_i|x_i)(r_i - \hat{q}(x_i, a_i))$$

$$+ \frac{1}{n} \sum_{i=1}^n \mathbb{E}_{a \sim \pi_\theta(a|x_i)}[\nabla_\theta \log \pi_\theta(a_i|x_i)\hat{q}(x_i, a)].$$

By using the regressed reward as a control variate, DR often reduces the variance of IS, while remaining unbiased under the same condition as IS. However, when the regression is inaccurate, the variance reduction is limited and DR often suffers from high variance when the action space is large (Saito & Joachims, 2022).

**POTEC (Saito et al., 2024).** To deal with the variance issue of DR, POTEC considers the clustering in the action space and decomposes the policy into two stages as follows.

$$\pi_\theta(a|x) = \pi_\theta^{1st}(c|x)\pi^{2nd}(a|x, c),$$

where $c$ indicates the cluster of the action $a$, which can be learned by applying an off-the-shelf clustering method to action embeddings. Using this decomposition, POTEC chooses clusters via a DR-style approach as follows, and chooses actions within a cluster via regression.

$$\nabla_\theta V(\pi_\theta) \approx \frac{1}{n} \sum_{i=1}^n \frac{\pi_\theta^{1st}(c(a_i)|x_i)}{\pi_0^{1st}(c(a_i)|x_i)} \nabla_\theta \log \pi_\theta^{1st}(c(a_i)|x_i)(r_i - \hat{q}(x_i, a_i))$$

$$+ \frac{1}{n} \sum_{i=1}^n \mathbb{E}_{a \sim \pi_\theta(a|x_i)}[\nabla_\theta \log \pi_\theta^{1st}(c(a_i)|x_i)\hat{q}(x_i, a)],$$

where $\pi_0^{1st}(c(a)|x) = \sum_{a' \in \mathcal{A}, c(a')=c(a)} \pi_0(a|x)$. The second-stage policy greedily chooses action as $\pi^{2nd}(a|x, c) = \mathbb{I}\{\hat{q}(x, a) = \arg\max_{a' \in \mathcal{A}, c(a')=c(a)} \hat{q}(x, a')\}$. By applying IS on the clustered action space, POTEC reduces the variance of naive IS. POTEC is also able to convert regression to a pair-wise regression within a cluster. However, especially when the relation between actions and rewards is complex, a good clustering is often hard to identify, and POTEC cannot take the rich information about generated sentences into account.

## D OMITTED PROOFS AND DERIVATIONS

This section provide proofs and derivations ommited in the main text.

### D.1 DERIVATION OF THE PG IN THE SENTENCE SPACE

We first derive $\nabla_\theta \log \pi_\theta(s|x) = \mathbb{E}_{a \sim \pi_\theta(a|x,s)}[\nabla_\theta \log \pi_\theta(a|x)]$.

$$\nabla_\theta \log \pi_\theta(s|x) = \frac{\nabla_\theta \pi_\theta(s|x)}{\pi_\theta(s|x)}$$

$$= \frac{\sum_{a \in \mathcal{A}} \nabla_\theta \pi_\theta(a|x) p_{LLM}(s|x, a)}{\pi_\theta(s|x)}$$

$$= \sum_{a \in \mathcal{A}} \frac{\nabla_\theta \pi_\theta(a|x)}{\pi_\theta(a|x)} \pi_\theta(a|x, s)$$

$$= \mathbb{E}_{\pi_\theta(a|x,s)}[\nabla_\theta \log \pi_\theta(a|x)]$$

Similarly, we also have $\nabla_\theta \log \pi_\theta(\phi(s)|x) = \mathbb{E}_{\pi_\theta(s'|x,\phi(s))}[\nabla_\theta \log \pi_\theta(s'|x)]$ thus $\nabla_\theta \log \pi_\theta(\phi(s)|x) = \mathbb{E}_{\pi_\theta(s'|x,\phi(s))}[\mathbb{E}_{\pi_\theta(a|x,s')}[\nabla_\theta \log \pi_\theta(a|x)]] = \mathbb{E}_{\pi_\theta(a|x,\phi(s))}[\nabla_\theta \log \pi_\theta(a|x)]$.

## D.2 DERIVATION OF THE WEIGHTED SCORE FUNCTION

We first show $w(\phi(s), x) = \mathbb{E}_{\pi_0(a|x,\phi(s))}[w(a, x)]$.

$$
\begin{aligned}
\mathbb{E}_{\pi_0(a|x,\phi(s))}[w(a, x)] &= \sum_{a\in\mathcal{A}} \pi_0(a|x, \phi(s)) \frac{\pi_\theta(a|x)}{\pi_0(a|x)} \\
&= \sum_{a\in\mathcal{A}} \pi_0(a|x, \phi(s)) \frac{\frac{\pi_\theta(a|x,\phi(s))\pi_\theta(\phi(s)|x)}{p_{\text{LLM}}(\phi(s)|x,a)}}{\frac{\pi_0(a|x,\phi(s))\pi_0(\phi(s)|x)}{p_{\text{LLM}}(\phi(s)|x,a)}} \\
&= \sum_{a\in\mathcal{A}} \pi_0(a|x, \phi(s)) \frac{\pi_\theta(a|x,\phi(s))}{\pi_0(a|x,\phi(s))} \frac{\pi_\theta(\phi(s)|x)}{\pi_0(\phi(s)|x)} \\
&= \mathbb{E}_{\pi_\theta(a|x,\phi(s))}[w(x, \phi(s))] \\
&= w(\phi(s), x)
\end{aligned}
$$

Next, using the above expression and that derived in Appendix D.1, we have

$$
\begin{aligned}
& w(\phi(s)|x)\nabla_\theta \log \pi_\theta(\phi(s)|x) \\
&= \frac{\pi_\theta(\phi(s)|x)}{\pi_0(\phi(s)|x)} \mathbb{E}_{\pi_\theta(s'|x,\phi(s))}[\mathbb{E}_{\pi_\theta(a|x,s')}[\nabla_\theta \log \pi_\theta(a|x)]] \\
&= \frac{\pi_\theta(\phi(s)|x)}{\pi_0(\phi(s)|x)} \int_{s'\in\mathcal{S}} \pi_\theta(s'|x, \phi(s)) \sum_{a\in\mathcal{A}} \pi_\theta(a|x, s')\nabla_\theta \log \pi_\theta(a|x) ds' \\
&= \frac{\pi_\theta(\phi(s)|x)}{\pi_0(\phi(s)|x)} \int_{s'\in\mathcal{S}} \frac{p_K(\phi(s)|x, s')\pi_\theta(s'|x)}{\pi_\theta(\phi(s)|x)} \sum_{a\in\mathcal{A}} \frac{p_{\text{LLM}}(s'|x, a)\pi_\theta(a|x)}{\pi_\theta(s'|x)}\nabla_\theta \log \pi_\theta(a|x) ds' \\
&= \frac{1}{\pi_0(\phi(s)|x)} \int_{s'\in\mathcal{S}} K(s, s'; x, \tau) \sum_{a\in\mathcal{A}} p_{\text{LLM}}(s'|x, a)\pi_\theta(a|x)\nabla_\theta \log \pi_\theta(a|x) ds' \\
&= \sum_{a\in\mathcal{A}} \pi_\theta(a|x) \int_{s'\in\mathcal{S}} p_{\text{LLM}}(s'|x, a) \frac{K(s, s'; x, \tau)}{\pi_0(\phi(s)|x)}\nabla_\theta \log \pi_\theta(a|x) ds' \\
&= \mathbb{E}_{\pi_\theta(a|x)p_{\text{LLM}}(s'|x,a)}\left[ \frac{K(s, s'; x, \tau)}{\pi_0(\phi(s)|x)}\nabla_\theta \log \pi_\theta(a|x) \right]
\end{aligned}
$$

where $p_K(\phi(s)|x, s') = K(s, s'; x, \tau)$.

## D.3 DERIVATION OF THE BIAS OF DSO (PROOFS OF THEOREM 1)

*Proof.* To derive the bias of DSO, we first decompose the expectation of DSO as follows.

$$
\begin{aligned}
& \mathbb{E}_{\mathcal{D}}[w(\phi(s'), x)\nabla_\theta \log \pi_\theta(\phi(s')|x)r] \\
&= \mathbb{E}_{\pi_0(s'|x)}[w(\phi(s'), x)\nabla_\theta \log \pi_\theta(\phi(s')|x)q(x, s')] \\
&= \mathbb{E}_{\pi_0(\phi(s)|x)\pi_0(s'|x,\phi(s))}[w(\phi(s'), x)\nabla_\theta \log \pi_\theta(\phi(s')|x)q(x, s')] \\
&= \mathbb{E}_{\pi_0(\phi(s)|x)\pi_0(s'|x,\phi(s))}[w(\phi(s'), x)\nabla_\theta \log \pi_\theta(\phi(s')|x)q(x, s')] \\
&\quad - \mathbb{E}_{\pi_0(\phi(s)|x)\pi_0(s'|x,\phi(s))}[w(\phi(s), x)\nabla_\theta \log \pi_\theta(\phi(s)|x)q(x, s')] \\
&\quad + \mathbb{E}_{\pi_0(\phi(s)|x)\pi_0(s'|x,\phi(s))}[w(\phi(s), x)\nabla_\theta \log \pi_\theta(\phi(s)|x)q(x, s')] \\
&= \mathbb{E}_{\pi_0(\phi(s)|x)\pi_0(s'|x,\phi(s))}[(w(\phi(s'), x)\nabla_\theta \log \pi_\theta(\phi(s')|x) - w(\phi(s), x)\nabla_\theta \log \pi_\theta(\phi(s)|x))q(x, s')] \\
&\quad + \mathbb{E}_{\pi_0(\phi(s)|x)\pi_0(s'|x,\phi(s))}[w(\phi(s), x)\nabla_\theta \log \pi_\theta(\phi(s)|x)q(x, s')] \\
&= \mathbb{E}_{\pi_0(\phi(s)|x)\pi_0(s'|x,\phi(s))}[\Delta_{(w\nabla_\theta)}(\phi(s'), \phi(s); x)q(x, s')] \\
&\quad + \mathbb{E}_{\pi_0(\phi(s)|x)}[w(\phi(s), x)\nabla_\theta \log \pi_\theta(\phi(s)|x)q^{\pi_0}(x, \phi(s))].
\end{aligned}
$$

Then, for the second term, we have

$$\mathbb{E}_{\pi_0(\phi(s)|x)}[w(\phi(s),x)\nabla_\theta \log \pi_\theta(\phi(s)|x)q^{\pi_0}(x,\phi(s))]$$

$$\mathbb{E}_{\pi_0(\phi(s)|x)}\left[\frac{\pi_\theta(\phi(s)|x)}{\pi_0(\phi(s)|x)}\nabla_\theta \log \pi_\theta(\phi(s)|x)q^{\pi_0}(x,\phi(s))\right]$$

$$= \sum_{\phi(s)\in\Phi(S)} \pi_0(\phi(s)|x)\frac{\pi_\theta(\phi(s)|x)}{\pi_0(\phi(s)|x)}\nabla_\theta \log \pi_\theta(\phi(s)|x)q^{\pi_0}(x,\phi(s))$$

$$= \sum_{\phi(s)\in\Phi(S)} \pi_\theta(\phi(s)|x)\nabla_\theta \log \pi_\theta(\phi(s)|x)q^{\pi_0}(x,\phi(s))$$

$$= \mathbb{E}_{\pi_\theta(\phi(s)|x)}[\nabla_\theta \log \pi_\theta(\phi(s)|x)q^{\pi_0}(x,\phi(s))].$$

Next, we also transform the true gradient in the sentence space as follows:

$$\mathbb{E}_{\pi_\theta(s'|x)}[\nabla_\theta \log \pi_\theta(s'|x)q(x,s')]$$

$$= \mathbb{E}_{\pi_\theta(\phi(s)|x)\pi_\theta(s'|x,\phi(s))}[\nabla_\theta \log \pi_\theta(s'|x)q(x,s')]$$

$$= \mathbb{E}_{\pi_\theta(\phi(s)|x)\pi_\theta(s'|x,\phi(s))}[\nabla_\theta \log \pi_\theta(s'|x)q(x,s')]$$

$$\quad - \mathbb{E}_{\pi_\theta(\phi(s)|x)\pi_\theta(s'|x,\phi(s))}[\nabla_\theta \log \pi_\theta(\phi(s)|x)q(x,s')]$$

$$\quad + \mathbb{E}_{\pi_\theta(\phi(s)|x)\pi_\theta(s'|x,\phi(s))}[\nabla_\theta \log \pi_\theta(\phi(s)|x)q(x,s')]$$

$$= \mathbb{E}_{\pi_\theta(\phi(s)|x)\pi_\theta(s'|x,\phi(s))}[(\nabla_\theta \log \pi_\theta(s'|x) - \nabla_\theta \log \pi_\theta(\phi(s)|x))q(x,s')]$$

$$\quad + \mathbb{E}_{\pi_\theta(\phi(s)|x)\pi_\theta(s'|x,\phi(s))}[\nabla_\theta \log \pi_\theta(\phi(s)|x)q(x,s')]$$

$$= \mathbb{E}_{\pi_\theta(\phi(s)|x)\pi_\theta(s'|x,\phi(s))}[\Delta_{(\nabla_\theta)}(s',\phi(s))q(x,s')]$$

$$\quad + \mathbb{E}_{\pi_\theta(\phi(s)|x)}[\nabla_\theta \log \pi_\theta(\phi(s)|x)q^{\pi_\theta}(x,\phi(s))].$$

Therefore, the bias is

$$\mathrm{Bias}((\widehat{\nabla_\theta V})_{\mathrm{DSO}})$$

$$= \mathbb{E}_{\mathcal{D}}[w(\phi(s'),x)\nabla_\theta \log \pi_\theta(\phi(s')|x)r] - \mathbb{E}_{\pi_\theta(s'|x)}[\nabla_\theta \log \pi_\theta(s'|x)q(x,s')]$$

$$= \mathbb{E}_{\pi_\theta(\phi(s)|x)}[\nabla_\theta \log \pi_\theta(\phi(s)|x)q^{\pi_0}(x,\phi(s))]$$

$$\quad + \mathbb{E}_{\pi_0(\phi(s)|x)\pi_0(s'|x,\phi(s))}[\Delta_{(w\nabla_\theta)}(\phi(s'),\phi(s);x)q(x,s')]$$

$$\quad - \mathbb{E}_{\pi_\theta(\phi(s)|x)}[\nabla_\theta \log \pi_\theta(\phi(s)|x)q^{\pi_\theta}(x,\phi(s))]$$

$$\quad - \mathbb{E}_{\pi_\theta(\phi(s)|x)\pi_\theta(s'|x,\phi(s))}[\Delta_{(\nabla_\theta)}(s',\phi(s))q(x,s')]$$

$$= \mathbb{E}_{\pi_\theta(\phi(s)|x)}[\nabla_\theta \log \pi_\theta(\phi(s)|x)(q^{\pi_0}(x,\phi(s)) - q^{\pi_\theta}(x,\phi(s)))]$$

$$\quad + \mathbb{E}_{\pi_0(\phi(s)|x)\pi_0(s'|x,\phi(s))}[\Delta_{(w\nabla_\theta)}(\phi(s'),\phi(s);x)q(x,s')]$$

$$\quad + \mathbb{E}_{\pi_\theta(\phi(s)|x)\pi_\theta(s'|x,\phi(s))}[\Delta_{(\nabla_\theta)}(\phi(s),s')q(x,s')]$$

$$= \mathbb{E}_{\pi_\theta(\phi(s)|x)}[\nabla_\theta \log \pi_\theta(\phi(s)|x)\Delta_q(\pi_\theta,\pi_0;\, x,\phi(s))]$$

$$\quad + \mathbb{E}_{\pi_0(\phi(s)|x)\pi_0(s'|x,\phi(s))}[\Delta_{(w\nabla_\theta)}(\phi(s'),\phi(s);x)q(x,s')]$$

$$\quad + \mathbb{E}_{\pi_\theta(\phi(s)|x)\pi_\theta(s'|x,\phi(s))}[\Delta_{(\nabla_\theta)}(\phi(s),s')q(x,s')].$$

where we define

$$\Delta_q(\pi_\theta,\pi_0;\, x,\phi(s)) := q^{\pi_\theta}(x,\phi(s)) - q^{\pi_0}(x,\phi(s)),$$

$$\Delta_{(w\nabla_\theta)}(\phi(s'),\phi(s);x) := w(\phi(s'),x)\nabla_\theta \log \pi_\theta(\phi(s')|x) - w(\phi(s),x)\nabla_\theta \log \pi_\theta(\phi(s)|x),$$

$$\Delta_{(\nabla_\theta)}(\phi(s),s') := \nabla_\theta \log \pi_\theta(s'|x) - \nabla_\theta \log \pi_\theta(\phi(s)|x).$$

$\square$

## D.4 DERIVATION OF THE VARIANCE OF DSO (PROOFS OF THEOREM 2)

*Proof.* From the total law of variance, we have

$$n\mathbb{V}((\widehat{\nabla_\theta V})_{\mathrm{DSO}}) = \mathbb{V}_{p(x)}(\mathbb{E}_{\mathcal{D}}[(\widehat{\nabla_\theta V})_{\mathrm{DSO}}|x])$$

$$\quad + \mathbb{E}_{p(x)}[\mathbb{V}_{\pi_0(s|x)}(w(\phi(s)|x)\nabla_\theta \log \pi_\theta(\phi(s)|x)q(x,s))]$$

$$\quad + \mathbb{E}_{p(x)\pi_0(s|x)}[(w(\phi(s)|x))^2(\nabla_\theta \log \pi_\theta(\phi(s)|x))^2\sigma^2(x,s)].$$

Because we have $w(\phi(s)|x) = \mathbb{E}_{\pi_0(a|x,\phi(s))}[w(x,a)]$ and $\textcolor{red}{\nabla_\theta \log \pi_\theta(\phi(s)|x) = \mathbb{E}_{\pi_0(a|x,\phi(s))}[\nabla_\theta \log \pi_\theta(a|x)]}$, the following holds.

$$\mathbb{V}_{\pi_0(a,s|x)}(w(a,x)) - \mathbb{V}_{\pi_0(a,s|x)}(w(\phi(s)|x)) = \mathbb{E}_{\pi_0(s|x)}[\mathbb{V}_{\pi_0(a|\phi(s)|x)}(w(a,x))]$$

$$\mathbb{V}_{\pi_0(a,s|x)}(\nabla_\theta \log \pi_\theta(s|x)) - \mathbb{V}_{\pi_0(a,s|x)}(\nabla_\theta \log \pi_\theta(a|x)) = \mathbb{E}_{\pi_0(s|x)}[\mathbb{V}_{\pi_0(a|\phi(s)|x)}(\nabla_\theta \log \pi_\theta(a|x))]$$

$\square$

# E FUTURE WORK: DISCUSSION ABOUT DR VARIANTS OF DSO

From the above theoretical analysis, the regression-based baseline required for a DR-style estimator like Dudík et al. (2011); Saito et al. (2023) should be

$$\mathbb{E}_{\pi_\theta(\phi(s)|x)}[\nabla_\theta \log \pi_\theta(\phi(s)|x)\hat{q}^{\pi_0}(x, \phi(s))]$$

in expectation to achieve the same degree of bias as IS. However, a way of computing such baselines is not trivial because estimating $\log \pi_\theta(\phi(s)|x)$ from data without applying importance sampling is challenging. Specifically, while it is possible to estimate the score function as follows, as we did in estimating the weighted score function,

$$\nabla_\theta \log \pi_\theta(\phi(s)|x) = \frac{\pi_0(\phi(s)|x)}{\pi_0(\phi(s)|x)}\nabla_\theta \log \pi_\theta(\phi(s)|x)$$

$$= \mathbb{E}_{(a,s')\sim\pi_0(a|x)p_{\text{LLM}}(s'|x,a)}\left[\frac{K(s,s';\,x,\tau)\nabla_\theta \log \pi_\theta(a|x)}{\pi_0(\phi(s)|x)}\right],$$

we need additional importance sampling in the regression-based baseline term, not only in the original IS term. Therefore, even though DR approaches often aim for a further variance reduction, this naive definition of DSO-hybrid does not reduce the variance of DSO-IS. Figuring out an efficient way of combining IS and regression would be a promising future work.

# F EXAMPLE USAGES OF OFFLINEPROMPTS

## F.1 SEMI-SYNTHETIC BENCHMARK WITH LANGUAGE GENERATION

Here, we provide example codes to streamline the OPL procedure using OfflinePrompts. While we focus on the movie description (semi-synthetic) benchmark in this section, a similar workflow is also applicable to the synthetic benchmark. Please also refer to additional example codes including those with the synthetic benchmark at: (double blind review).

### F.1.1 SETTING UP A SEMI-SYNTHETIC SIMULATION

To set up the default movie description benchmark, users can follow the codes in Code snippet 1. The default datasets, candidate prompts, and finetuned parameters are stored in `src/dataset/assets/` in the OfflinePrompts repository.

To customize the benchmark setting, it is also possible to use configurable sub-modules: `ContextQueryLoader`, `CandidateActionsLoader`, `FrozenLLM`, and `RewardSimulator`. Specifically, users can first create customized instances of these submodules and then pass them to `SemiSyntheticDataset` as exemplified in Code snippet 2-5.

### F.1.2 LOGGING POLICY

After setting up the simulator, the next step is to define a logging policy to collect logged feedback. We describe the procedure in Code snippets 6 and 7. Specifically, in Code snippet 6, we first fit the dimension reduction model to obtain low dimensional embeddings of query, prompt, and sentence. These encoders are used across various models, e.g., to define the logging policy and to define a reward preditor, etc. Then, Code snippet 7 describes how to define a softmax logging policy. In the example code, we first train a regression model used in the logging policy and then pass it to the softmax policy class.

### F.1.3 DATA COLLECTION AND REGRESSIONS

Once defining a logging policy, we collect logged data as shown in Code snippets 8. The outputs, including `logged_feedback` and `meta_data`, contain the following keys.

- `logged_feedback`:
  { user_id, item_id, context, query, action, action_choice_probability*, sentence, expected_reward*, reward }
- `meta_data`*:
  { size, reward_type, reward_std, action_list }

Note that the keys with an asterisk (*) are optional outputs, and action is returned by index. `reward_type` indicates whether the reward is binary or continuous, and `action_list` contains the list of candidate prompts, corresponding to each action index.

After obtaining the logged data, we regress the reward and train a logging marginal density model as described in Code snippets 8 and 9. `prompt_reward_predictor` is used by naive PG and two-stage PG, while `sentence_reward_predictor` and `marginal_density_model` are used by DSO.

### F.1.4 (ONLINE POLICY GRADIENT)

In OPL experiments, we often use the performance of online policy gradient as a baseline. To learn a policy online, we can run online policy gradient as shown in Code snippet 10.

### F.1.5 SINGLE STAGE POLICY GRADIENTS

Code snippet 11 shows the example codes to run naive PGs, including regression-based, IS-based, and hybrid ones. The procedure consists of only 3 steps: (1) define a policy, (2) then setup a learner class (`PolicyLearner`), and (3) call one of the policy gradient methods. As seen in the example code, all policy gradient methods can be called in similar formats. Researchers can also implement their own policy gradient methods in a similar way.

### F.1.6 DIRECT SENTENCE OFF-POLICY GRADIENT (DSO)

DSO can also be run in a very similar way as the naive policy gradient. As exemplified in Code snippet 12, the key difference is that DSO uses `KernelPolicyLearner`, `logging_marginal_density_model`, and `sentence_reward_predictor`. Only the IS-based policy gradient is implemented for DSO.

### F.2 (ONLINE) PERFORMANCE EVALUATION

Finally, after learning a policy, we test its performance through online interaction. This can be done in a single line of code, as shown in Code snippet 13.

We also provide additional quickstart examples at: (double blind review)

```python
from src.dataset import SemiSyntheticDataset
dataset = SemiSyntheticDataset(
    path_to_user_embeddings="assets/movielens_naive_cf_user_embeddings",
    path_to_queries="assets/movielens_query.csv",
    path_to_candidate_prompts="assets/movielens_benchmark_prompts.csv",
    path_to_finetuned_params= "assets/movielens_distilbert_reward_simulator.pt",
    random_state=12345,
)
```

Code Snippet 1: Setting up the default benchmark environment

```python
from src.dataset import (
    ContextQueryLoader,
    CandidateActionsLoader,
)

# load contexts and queries
context_query_loader = DefaultContextQueryLoader(
    path_to_user_embeddings="assets/movielens_naive_cf_user_embeddings",
    path_to_queries="assets/movielens_query.csv",
    device=self.device,
    random_state=self.random_state,
            )

# load candidate prompts
candidate_actions_loader = CandidateActionsLoader(
    n_actions=1000,
    path_to_candidate_prompts="assets/movielens_benchmark_prompts.csv",
    random_state=self.random_state,
)
```

Code Snippet 2: Customizing the context, query, candidate actions loader

```python
from src.dataset import AutoFrozenLLM
from transformers import AutoModelForCausalLM, AutoTokenizer

# load frozen llm
frozen_llm_tokenizer = AutoTokenizer.from_pretrained(
    "mistralai/Mistral-7B-Instruct-v0.2",
    truncation=True,
    do_lower_case=True,
    use_fast=True,
)
frozen_llm_model = AutoModelForCausalLM.from_pretrained(
    "mistralai/Mistral-7B-Instruct-v0.2",
)
frozen_llm_tokenizer_kwargs = {
    "add_special_tokens": True,
    "padding": True,
    "truncation": True,
    "max_length": 20,
    "return_tensors": "pt",
}
self.frozen_llm_prompt_formatter = MovielensPromptFormatter(
        tokenizer=frozen_llm_tokenizer,
        tokenizer_kwargs=frozen_llm_tokenizer_kwargs,
        device="cuda",
    )
pattern = r"Broadly describe in a sentence the genres of the movie without
    including the name or any specifics of.*?\n\n"

frozen_llm_tokenizer.add_special_tokens({"pad_token": "[PAD]"})
frozen_llm_model.resize_token_embeddings(len(frozen_llm_tokenizer))
frozen_llm_model.to("cuda")

frozen_llm = AutoFrozenLLM(
    prompt_formatter=frozen_llm_prompt_formatter,
    model=frozen_llm_model,
    tokenizer=frozen_llm_tokenizer,
    tokenizer_kwargs=frozen_llm_tokenizer_kwargs,
    pattern=pattern,
    device="cuda",
    random_state=12345,
)
```

Code Snippet 3: Customizing the frozen LLM

```python
from src.dataset import TransformerRewardSimulator
from transformers import AutoModelForCausalLM, AutoTokenizer

# load reward simulator
reward_simulator_tokenizer = AutoTokenizer.from_pretrained(
    "distilbert-base-uncased",
    truncation=True,
    do_lower_case=True,
    use_fast=True,
)
reward_simulator_base_model = AutoModel.from_pretrained(
    "distilbert-base-uncased",
)
reward_simulator_tokenizer_kwargs = {
    "add_special_tokens": True,
    "padding": True,
    "truncation": True,
    "max_length": 20,
    "return_tensors": "pt",
}

reward_simulator_tokenizer.add_special_tokens({"pad_token": "[PAD]"})
reward_simulator_base_model.resize_token_embeddings(
    len(reward_simulator_tokenizer)
)
reward_simulator_base_model.to("cuda")

reward_simulator = TransformerRewardSimulator(
    n_users=context_query_loader.n_users,
    n_items=context_query_loader.n_queries,
    base_model=reward_simulator_base_model,
    tokenizer=reward_simulator_tokenizer,
    tokenizer_kwargs=reward_simulator_tokenizer_kwargs,
    device="cuda",
    random_state=12345,
)
reward_simulator.load_state_dict(
    torch.load("assets/movielens_distilbert_reward_simulator.pt")
)
```

Code Snippet 4: Customizing the reward simulator

```python
# create a custom environment with customized modules
dataset = SemiSyntheticDataset(
    context_query_loader=context_query_loader,
    candidate_actions_loader=candidate_actions_loader,
    frozen_llm=frozen_llm,
    reward_simulator=reward_simulator,
    frozen_llm_prompt_formatter=frozen_llm_prompt_formatter,
    reward_type="binary",
    device="cuda",
    random_state=12345,
)
```

Code Snippet 5: Combining the customized modules to define the custom dataset

```python
from src.policy import TransformerEncoder
from src.policy import UniformRandomPolicy

# collect logged data to fit the encoder
uniform_policy = UniformRandomPolicy(
    action_list=dataset.action_list,
    device="cuda",
    random_state=12345,
)
logged_feedback_for_pretraining = dataset.sample_dataset(
    policy=uniform_policy, n_samples=10000,
)
query = logged_feedback_for_pretraining["query"]
sentence = logged_feedback_for_pretraining["sentence"]
prompt = dataset.action_list

# define and fit encoders
query_encoder = TransformerEncoder(
    dim_emb=10,
    device="cuda",
    random_state=12345,
)
prompt_encoder = TransformerEncoder(
    dim_emb=10,
    device="cuda",
    random_state=12345,
)
sentence_encoder = TransformerEncoder(
    dim_emb=10,
    device="cuda",
    random_state=12345,
)
# applying dimension reduction
query_encoder.fit_pca(query)
prompt_encoder.fit_pca(prompt)
sentence_encoder.fit_pca(sentence)
```

Code Snippet 6: Fitting encoder

```
from src.opl import PromptRewardLearner
from src.policy import PromptRewardPredictor
from src.policy import SoftmaxPolicy, UniformRandomPolicy

# collect data for regressing the logging reward predictor
uniform_policy = UniformRandomPolicy(
    action_list=dataset.action_list,
    device="cuda",
    random_state=12345,
)
logged_feedback_for_pretraining = dataset.sample_dataset(
    policy=uniform_policy, n_samples=10000,
)

# train a reward predictor to define the logging policy
prompt_reward_predictor = PromptRewardPredictor(
    dim_context=dataset.dim_context,
    action_list=dataset.action_list,
    query_encoder=query_encoder,
    prompt_encoder=prompt_encoder,
    device="cuda",
    random_state=12345,
)
prompt_reward_learner = PromptRewardLearner(
    model=prompt_reward_predictor,
    action_list=dataset.action_list,
    frozen_llm=dataset.frozen_llm,
    optimizer_kwargs={"lr": 1e-4, "weight_decay": 0.0},
    env=dataset,
    random_state=12345,
)
logging_prompt_reward_predictor = prompt_reward_learner.offline_training(
    logged_feedback=logged_feedback_for_pretraining,
    random_state=12345,
)

# define logging policy
logging_policy = SoftmaxPolicy(
    action_list=dataset.action_list,
    base_model=logging_prompt_reward_predictor,
    beta=1.0, # inversed temperature
    device="cuda",
    random_state=12345,
)
```

Code Snippet 7: Training the logging reward predictor and defining the logging policy

```python
from src.opl import SentenceRewardLearner, PromptRewardLearner
from src.policy import SentenceRewardPredictor, PromptRewardPredictor

# collect logged dataset
logged_feedback = dataset.sample_dataset(
    policy=logging_policy,
    n_samples=10000,
)

# train regression models
sentence_reward_predictor = SentenceRewardPredictor(
    dim_context=dataset.dim_context,
    frozen_llm=dataset.frozen_llm,
    query_encoder=query_encoder,
    sentence_encoder=sentence_encoder,
    device="cuda",
    random_state=12345,
)
sentence_reward_learner = SentenceRewardLearner(
    action_list=dataset.action_list,
    model=sentence_reward_predictor,
    frozen_llm=dataset.frozen_llm,
    optimizer_kwargs={"lr": 1e-4, "weight_decay": 0.0},
    random_state=12345,
)
sentence_reward_predictor = sentence_reward_learner.offline_training(
    logged_feedback=logged_feedback,
)
prompt_reward_predictor = PromptRewardPredictor(
    dim_context=dataset.dim_context,
    action_list=dataset.action_list,
    query_encoder=query_encoder,
    prompt_encoder=prompt_encoder,
    device="cuda",
    random_state=12345,
)
prompt_reward_learner = PromptRewardLearner(
    model=prompt_reward_predictor,
    action_list=dataset.action_list,
    frozen_llm=dataset.frozen_llm,
    query_encoder=query_encoder,
    prompt_encoder=prompt_encoder,
    optimizer_kwargs={"lr": 1e-4, "weight_decay": 0.0},
    env=dataset,
    random_state=12345,
)
prompt_reward_predictor = prompt_reward_learner.offline_training(
    logged_feedback=logged_feedback,
)
```

Code Snippet 8: Collecting logged data and regressing rewards

```python
from src.opl import MarginalDensityLearner
from src.policy import KernelMarginalDensityEstimator
from src.utils import gaussian_kernel

# learning a marginal density model
kernel_marginal_estimator = KernelMarginalDensityEstimator(
    action_list=dataset.action_list,
    dim_context=dataset.dim_context,
    frozen_llm=dataset.frozen_llm,
    query_encoder=query_encoder,
    sentence_encoder=sentence_encoder,
    kernel_function=gaussian_kernel,
    kernel_kwargs={"tau": 1.0}, # bandwidth
    device="cuda",
    random_state=12345,
)
marginal_density_learner = MarginalDensityLearner(
    model=kernel_marginal_estimator,
    action_list=dataset.action_list,
    frozen_llm=dataset.frozen_llm,
    optimizer_kwargs={"lr": 1e-4, "weight_decay": 0.0},
)
kernel_marginal_estimator = marginal_density_learner.simulation_training(
    logged_feedback=logged_feedback,
)
```

Code Snippet 9: Training a logging marginal density model (used by DSO)

```python
from src.opl import PolicyLearner
from src.policy import PromptPolicy

policy = PromptPolicy(
    n_actions=dataset.n_actions,
    dim_context=dataset.dim_context,
    query_encoder=query_encoder,
    device="cuda",
    random_state=12345,
)
policy_learner = PolicyLearner(
    model=policy,
    action_list=dataset.action_list,
    prompt_reward_predictor=prompt_reward_predictor,
    query_encoder=query_encoder,
    sentence_encoder=sentence_encoder,
    optimizer_kwargs={"lr": 5e-4, "weight_decay": 0.0},
    env=dataset,
    random_state=12345,
)
policy = policy_learner.online_policy_gradient(
    n_epochs=10000,
)
```

Code Snippet 10: Online policy gradient

```python
from src.opl import PolicyLearner
from src.policy import PromptPolicy

policy = PromptPolicy(
    n_actions=dataset.n_actions,
    dim_context=dataset.dim_context,
    query_encoder=query_encoder,
    device="cuda",
    random_state=12345,
)
policy_learner = PolicyLearner(
    model=policy,
    action_list=dataset.action_list,
    prompt_reward_predictor=prompt_reward_predictor,
    query_encoder=query_encoder,
    sentence_encoder=sentence_encoder,
    optimizer_kwargs={"lr": 5e-4, "weight_decay": 0.0},
    env=dataset,
    random_state=12345,
)
# regression-based
policy = policy_learner.model_based_policy_gradient(
    logged_feedback=logged_feedback,
    n_epochs=10000,
)
# IS-based
policy = policy_learner.importance_sampling_based_policy_gradient(
    logged_feedback=logged_feedback,
    n_epochs=10000,
)
# hybrid
policy = policy_learner.hybrid_policy_gradient(
    logged_feedback=logged_feedback,
    n_epochs=10000,
)
```

Code Snippet 11: OPL with naive policy gradients

```python
from src.opl import PolicyLearner
from src.policy import PromptPolicy

policy = PromptPolicy(
    n_actions=dataset.n_actions,
    dim_context=dataset.dim_context,
    query_encoder=query_encoder,
    device="cuda",
    random_state=12345,
)
kernel_policy_learner = KernelPolicyLearner(
    model=policy,
    action_list=dataset.action_list,
    kernel_marginal_estimator=kernel_marginal_estimator,
    sentence_reward_predictor=sentence_reward_predictor,
    frozen_llm=dataset.frozen_llm,
    query_encoder=query_encoder,
    sentence_encoder=sentence_encoder,
    optimizer_kwargs={"lr": 5e-4, "weight_decay": 0.0},
    env=dataset,
    random_state=12345,
)
# DSO
policy = kernel_policy_learner.importance_sampling_based_policy_gradient(
    logged_feedback=logged_feedback,
    n_epochs=10000,
)
```

Code Snippet 12: OPL with DSO (the proposed method)

```python
policy_value = dataset.calc_expected_policy_value(
    policy=policy,
    n_samples_to_approximate=10000,
)
```

Code Snippet 13: Evaluating the policy performance online

