# OpenReview forum: "Prompt Optimization with Logged Bandit Data"
_ICLR.cc/2025/Conference — Submitted to ICLR 2025_

### Official Review · Reviewer_H6Mu · 2024-10-17

**Soundness:** 3
**Presentation:** 3
**Contribution:** 3
**Rating:** 8
**Confidence:** 2

**Summary:**

The paper introduces a new method for offline prompt policy learning for LLMs. The main challenge in this setting is the distribution shift between the logged data and the target data. Importance sampling can correct the distribution shift but only at the cost of potentially very high variance. The key idea behind the new method is to exploit similarity relations between sentences to reduce the variance. The bias-variance trade-off of the new method is analyzed theoretically and the method is tested on synthetic data and a LLM movie description task.

**Strengths:**

* The method is well-motivated and the theoretical analysis supports the desired variance reduction. Intuition for the analysis is provided.
* Ablations w.r.t to differences in the setting (dataset size, number of actions, reward noise) and w.r.t to the hyperparameters (kernel type, kernel bandwidth) of the method are carried out.
* Plan to open-source a benchmark for offline prompt policy learning

**Weaknesses:**

* Figure 6: there are 5 bars for each method. I was/am a bit confused about what the difference between these bars is. For now, I assume these are the results from the 5 random seeds, ordered by performance. But I think it would be good to have a label for this or mention it in the Figure caption.
* Literature on contextual bandits/kernelized bandits is left out.
* The performance gain (in particular compared to regression) seems much stronger in the synthetic setting than in the full-LLM experiment.

**Questions:**

See weaknesses.

**Details Of Ethics Concerns:**

I am not an expert on this, but I suspect that increasing personalization can also have potentially harmful social consequences (e.g. by reinforcing bubbles). On the other hand, I don't see an immediately greater risk than for other personalization methods that already exist and are widely accepted. So, I guess, it's fine.

---

> ### Author Response · Authors · 2024-11-17
> **Responses to the review**
>
> We would like to thank the reviewer for valuable feedback and the acknowledgment of the contributions. We respond to the key comments and questions below.
>
> > **Comment 1. Figure 6**
>
> Thank you for the valuable feedback. We will provide additional descriptions in the figure caption.
>
>
> > **Comment 2. Literature on contextual bandits/kernelized bandits.**
>
> Thank you for the great suggestion. We agree that the discussion of online kernelized bandits would be helpful, and we will incorporate it in the final version of the paper.
>
>
> > **Comment 3. Performance gain over regression.** “The performance gain (in particular compared to regression) seems much stronger in the synthetic setting than in the full-LLM experiment.”
>
> The main reason for the difference comes from the difference of reward noises in the synthetic and full-LLM experiments. In Figure 4 (synthetic experiment), we observe that the compared methods are competitive with the proposed method when the reward noise is zero, while the proposed method shows more promising results over others with the increase of reward noises. In contrast, the reward noise in the full-LLM experiment turned out (relatively) small, therefore, the performance gain is stronger in most of the synthetic experiment settings.

---

> > ### Comment · Reviewer_H6Mu · 2024-11-25
> >
> > Thank you for the reply. Having read the other reviews and rebuttals, I keep my good score.

---

### Official Review · Reviewer_MsZw · 2024-11-05

**Soundness:** 4
**Presentation:** 3
**Contribution:** 4
**Rating:** 8
**Confidence:** 4

**Summary:**

The paper presents Direct Sentence Off-policy gradient (DSO) for optimizing large language model (LLM) pipelines using logged user feedback such as clicks. DSO addresses the challenges of high variance and bias in policy gradient estimation by leveraging the similarity among generated sentences. The paper provides theoretical analysis on the source of bias and variance reduction of DSO. Experiments on both synthetic environment and a proposed benchmark (OfflinePrompts based on MovieLens-10M) demonstrate the effectiveness of this method. OfflinePrompts is a new benchmark suite,to demonstrate DSO's effectiveness in generating personalized movie descriptions. This is an additional contribution of the paper by providing a practical solution for leveraging naturally logged feedback for prompt policy optimization in language generation tasks.

**Strengths:**

- The algorithm DSO motivated by utilizing the information behind the sentence embedding is generally sound.
- The theoretically anslysis highlights the benefit of such algorithimic designs by indicating the source of bias and variance of such algorithms.
- The introduction of the OfflinePrompts benchmark suite is a valuable resource for the research community, facilitating further development and testing of off-policy learning methods for language generation

**Weaknesses:**

The experiments for real-world validation is insufficient. (Indeed, we lack good benchmarks for this task.) How well does the real-world performance align with the score/reward in the simulated environment (OfflinePrompts)? I found Figure 11 in the appendix indicates the positive correlation between the simulated rewards and the click feedback from users. Is there other statistics (such as the accuracy)? I am curious on the click rate improvement using the policy trained by DSO in real-world settings.

**Questions:**

How well the sythetic environments represent the real case? I note that there are some gaps between the sythetic environments and the target task. For example, reward is real-valued in synthetic case but it is binary in the real case (click or not); the policy is parameterized by an estimated reward function in the sythetic case.

---

> ### Author Response · Authors · 2024-11-17
> **Responses to the review**
>
> We would like to thank the reviewer for valuable feedback and the acknowledgment of the contributions. We respond to the key comments and questions below.
>
> > **Comment and question. Alignment between simulation and real-world env**: “How well does the real-world performance align with the score/reward in the simulated environment (OfflinePrompts)?” “Is there other statistics (such as the accuracy)?” “How well the sythetic environments represent the real case?”
>
> Thank you for the great point. We fit the sentence-based reward simulator using the MSE loss following the standard training protocol of collaborative-filtering (CF) models and confirm that the trained model achieves a competitive loss with the conventional CF model using user and item ID embeddings (MSE was around 0.22). Therefore, we believe that the trained simulator is sufficiently aligned with the real-world dataset.
>
> Although we did our best in replicating realistic situations, we should also note that there are some inevitable sim2real gaps in the benchmark environment, as it is impossible to completely mimic the real-world situation without accessing actual services. However, this is a common open challenge in bandits and RL settings, and still, many RL benchmarks including gymnasium (https://gymnasium.farama.org/index.html) provide useful insights in research papers. So is ours, and we believe our benchmark shows the proof-of-concepts. Nonetheless, we acknowledge the importance of your point.

---

> > ### Comment · Reviewer_MsZw · 2024-11-18
> >
> > Thanks for the additional information.

---

### Official Review · Reviewer_Qx8z · 2024-11-05

**Soundness:** 2
**Presentation:** 2
**Contribution:** 2
**Rating:** 3
**Confidence:** 3

**Summary:**

This paper addresses prompting policy optimization for large language model (LLM) pipelines by leveraging logged user feedback, such as clicks, to generate personalized sentences. The authors propose a novel method called Direct Sentence Off-policy gradient (DSO), which uses similarity between generated sentences to estimate the policy gradient. While this approach relies on importance sampling, it can reduce the variance of importance weights by treating them in a sentence space rather than the prompt space. Experiments on a synthetic task and an LLM-based task for personalized movie descriptions are shown to claim the effectiveness of the proposed DSO method.

**Strengths:**

* Using similarity in the generated sentence space to control the bias-variance tradeoff through importance weights is an interesting approach.
* The paper evaluates the proposed method on two types of tasks, synthetic and LLM-based tasks, demonstrating applicability in varied settings.
* Theoretical analysis provides insights into the characteristics of DSO, although some detailed proofs could not be fully verified by the reviewer.

**Weaknesses:**

**Lack of clarity in algorithmic steps**:
The specific steps for implementing the algorithm are unclear. It seems that gradient estimation would require sampling from both the prompt policy and the LLM. If this understanding is correct, how many samples would need to be generated per data point? Should this match the $m$  samples used to estimate $\pi_0(\phi(s_i)|x_i)$?

**Notation abuse and lack of clarity in definitions**:
This paper has some notation abuse, which leads to ambiguity. For example, the authors introduce $\pi_\theta(a|x, s)$ or $\pi_\theta(a|x, \phi(s))$ as a conditional distribution over prompts given the generated sentence $s$ in  Section 4 and Appendix D.1. However, this is problematic because $\pi_\theta$ is originally defined as a prompt selection policy and should not depend on $s$, which the LLM generates after selecting $a$. Additionally, while the expressions are somewhat interpretable, there is a lack of consistency in function arguments throughout the paper. For instance, $\pi_\theta(s|x)$ is used without explanation as $\sum_a \pi_\theta(a|x) p_{LLM}(s|x, a)$. To improve clarity, the authors should avoid redefining $\pi_\theta$ with different inputs and instead provide explicit auxiliary definitions where needed, along with a rationale for introducing these conditional probabilities.

**Unpractical setting  in Full-LLM Experiment with MovieLens**:
The LLM-based experiment in Section 7 lacks realistic user personalization. As shown in Figures 10 and 12, the prompt policy reduces user information to a single word (from a set of only 1000 words) before feeding it to the LLM. This simplistic representation raises concerns about whether the Full-LLM experiment setup can effectively capture real-world personalization. Without a richer prompt (e.g., short sentences) to convey nuanced user information, it is unclear if this approach offers any advantage over simply passing user attributes directly to the LLM. Consequently, this setup might be better categorized as a toy task rather than a realistic evaluation of the proposed method's applicability in real-world tasks.

**Concerns regarding the formulation of baseline approaches**:
The problem formulation in this work is novel; however, applying existing methods, particularly the regression approach, seems overly naive for this setup. Since the LLM that generates $s$ is available in this setup, it would be more appropriate for the reward predictor to take $(x, s)$ as input instead of $(x, a)$. Otherwise, the reward predictor would have to learn the LLM's inherent randomness (noise), which seems inefficient. Using $(x, s)$ would allow the reward predictor to avoid this redundancy and better capture the generated sentence features. A Nadaraya-Watson kernel regression (using the same kernel as in DSO) or a neural model like DistilBERT could be employed as the reward predictor to improve adaptability. In connection with the above, in the numerical experiments, using $(x, a)$ as the reward predictor's input in the regression approach may be unfair as a baseline comparison against DSO. DSO leverages (multiple) generated sentence(s) $s’$ for each context $x$ sampled from $\pi_\theta$ and the LLM. Thus, any observed performance gap between DSO and the regression approach may simply be due to this difference in formulation rather than any inherent advantage of DSO.

**Organization of the paper**:
The structure of the paper could be improved. For instance, details of the synthetic experiment setting and Section 4.2 (not cited in the main text) could be moved to the appendix, as these sections may be of lower priority for understanding the main contributions. Shifting these sections would allow more space for core elements like detailed algorithmic steps, problem setup, and full LLM experiment details in the main text.

**Questions:**

**Figure 6 Interpretation**:
It seems that each bar in Figure 6 represents the results across 5 random seeds. Given the variation across seeds, can we still conclude that the proposed method (DSO) consistently outperforms the regression baseline? The performance between DSO and regression appears similar when accounting for this variability.

**Minor comments**
* Line 391: $\sigma_o$ should be $\sigma_s$?
* Line 989: MSE loss should be $\sum_{i=1}^{n} (r_i - \hat{q}(x_i, a_i))^2$ instead of $\sum_{i=1}^{n} (r_i - \hat{q}(x_i, a_i))$.
* Line 1075: $\nabla_{\theta} \pi_{\theta}$ should be $\nabla_{\theta} \log \pi_{\theta}$?
* In Section 3.1, the classification of "conventional approaches" into "regression-based methods" and "importance sampling (IS)" feels somewhat unclear. It may be more intuitive to categorize these as "reward predictor-based approaches" and "reward predictor-free approaches." This distinction clarifies that IS methods directly use observed rewards, whereas regression-based methods estimate rewards across all actions.

---

> ### Author Response · Authors · 2024-11-17
> **Responses to the review (1/4)**
>
> We would like to thank the reviewer for valuable feedback and for the comments and questions for additional clarity. Below, we address the key comments and questions.
>
> > **Comment 1-1. Unpractical setting in Full-LLM Experiment with MovieLens**: “The LLM-based experiment in Section 7 lacks realistic user personalization. .. the prompt policy reduces user information to a single word (from a set of only 1000 words) .. Without a richer prompt (e.g., short sentences)” “it is unclear if this approach offers any advantage over simply passing user attributes directly to the LLM”
>
> Thank you for providing the opportunity to resolve these concerns. For the first set of comments regarding the choice of candidate prompts, we consider that the current setting is sufficient for the following reasons. First, **our framework is agnostic to the length of the prompts and does not require any changes to handle more complex prompts such as short sentences**. Second, the key measure of complexity for our approach is the number of potential prompts, and our experiments show that we can robustly handle a large number of prompts.
>
> For the second question, note that finding the most effective prompt can be quite different from adding more user attributes. This is particularly true in most **practical recommender systems**, where users are represented as a vector of high-dimensional embedding features with no interpretable meaning to the LLM. Even if our task was heavily influenced by the practicality of the experiment given the available data, this makes our task practically relevant as it corresponds to searching for the most effective text expression (prompt) given the user attributes. Moreover, in applications like **educational chatbots**, one may aim to identify good prompts to generate motivational comments for individual users. However, even when we have access to the raw user attributes or historical interaction data, it is hard to generate a suitable sentence for each user by merely inputting the raw data. Nonetheless, prompt policy learning allows us to identify good prompts that can easily steer the sentence generation, without fine-tuning LLMs. For these reasons, **learning a prompt policy does have significant advantages, especially when it corresponds to searching for the right text expression of user attributes**.
>
>
> > **Comment 1-2. Concerns regarding the formulation of baseline approaches**: “predictor to take $(x, s)$ as input instead of $(x, a)$, etc.”
>
> Thank you for the thoughtful questions, and we would be happy to resolve your concerns. First, the reason we used $\hat{q}(x, a)$ in our experiment is simply to make sure that all regression-involved baselines, i.e., regression-based PG, DR, and POTEC, share the same regression model. We did not consider the use of $\hat{q}(x, a)$ as an unfair treatment due to the following reasons.
>
> First, in our full-LLM experiment, we have verified that there was no variance in the sentence generation, i.e., **prompt and sentence have one-to-one correspondence**. This is because we used (deterministic) beam search, which is the default implementation of huggingface. Therefore, statistically we have $q(x, a) = q(x, s(x, a))$, and **we do not have the concern of “the reward predictor would have to learn the LLM's inherent randomness (noise)”** as the reviewer mentions.
>
> Second, when inputting the information about prompts to the neural network, **we actually input the embeddings of the sentence generated by Mistral-7B with the following instruction to the neural network as the prompt embeddings**: *"Associate the word - [prompt] - in the context of movie"*, which should be rich enough features of prompts. We indeed lacked these implementation details in the initial manuscript, and we will make sure to include them in the Appendix. We appreciate the reviewer’s useful feedback.

---

> > ### Comment · Reviewer_Qx8z · 2024-11-21
> >
> > Thank you for the response and for addressing my concerns.
> >
> > > Comment 1-1. Unpractical setting in Full-LLM Experiment with MovieLens:
> >
> > While I appreciate the clarification, the claims in the response would benefit from being directly supported by relevant references. In particular, I feel there may be differences or connections worth discussing with the following works:
> >
> > * Chat-REC: Towards Interactive and Explainable LLMs-Augmented Recommender Systems (https://arxiv.org/abs/2303.14524)
> > * PALR: Personalization Aware LLMs for Recommendation (https://arxiv.org/abs/2305.07622)
> >
> > Additionally, could the author(s) clarify whether the user information x in the experiments is represented as embedding features or textual data? If textual data is used, comparing the performance of directly inputting user information into the LLM would be helpful versus using the prompt policy framework to support the claims further.
> >
> > > Comment 1-2. Concerns regarding the formulation of baseline approaches:
> >
> > I appreciate the explanation. However, my initial concern remains: even without LLM randomness, the reward function fundamentally depends on the generated sentence $s$ or $\phi(s)$ rather than the prompt $a$. This suggests that using $q(x, s)$ or $q(x, \phi(s))$ might be a more appropriate formulation.
> >
> > Additionally, it would strengthen the paper to include experiments where baselines also use $q(x, \phi(s))$  to ensure that the performance improvements of DSO are not primarily due to differences in representation. Comparing these results would provide a clearer understanding of the actual advantages of the proposed method.

---

> > > ### Author Response · Authors · 2024-11-22
> > >
> > > Thank you for the thoughtful comments and suggestions for the related papers. We will include them in the final version of the paper.
> > >
> > > > Additionally, could the author(s) clarify whether the user information x in the experiments is represented as embedding features or textual data?
> > >
> > > Absolutely. We used *vectorial embeddings* learned by a naive collaborative filtering (CF) model as the user information $x$ in our experiment. Specifically, we trained a model to predict the (binarized) user rating using only user ID embeddings and item ID embeddings based on the MovieLens dataset, and exploit the user embeddings learned from this training procedure. (Note that these user embeddings are different from those used in the reward simulator, as the reward simulator employs a different CF model that is based on the sentence encoder.)
> > >
> > >
> > > > This suggests that using $q(x, s)$ or $q(x, \phi(s))$ might be a more appropriate formulation. .. Additionally, it would strengthen the paper to include experiments where baselines also use $\hat{q}(x, \phi(s))$ to ensure that the performance improvements of DSO are not primarily due to differences in representation.
> > >
> > > Thank you for clarifying these points. Based on your review, we additionally ran the regression-based PG with $\hat{q}(x, s)$ in the full-LLM experiment. We confirmed that the performance statistics did not change so much between $\hat{q}(x, a)$ and $\hat{q}(x, s)$ as follows, suggesting that the performance difference is *not* due to the difference between the representation of $a$ and $s$.
> > >
> > > Performances of 5 random seeds in the descending order:
> > >
> > > $\hat{q}(x, a)$: 0.208, 0.207, 0.107, 0.07, -0.00
> > >
> > > $\hat{q}(x, s)$: 0.211, 0.172, 0.110, 0.07, -0.01
> > >
> > > ---
> > >
> > > We are grateful for your insights and would be happy to address any further questions or concerns.

---

> > > > ### Comment · Reviewer_Qx8z · 2024-11-25
> > > >
> > > > Thank you for conducting the additional experiments. However, in the full-LLM experiments, the regression approach already performs similarly to the proposed DSO method, as acknowledged by the authors:
> > > >
> > > > > > Question 1. Figure 6 Interpretation: "The performance between DSO and regression appears similar when accounting for this variability."
> > > >
> > > > > We agree with the reviewer's observation that regression also works well in the full-LLM experiments; DSO and regression perform better than other IS-based baselines.  ...
> > > >
> > > > Thus, this additional experiment confirms that the regression approach remains on par with DSO even when modified to use q(x, s). This result does not provide new evidence to support the superiority of DSO.
> > > >
> > > > Moreover, given the variance in the reported results (as seen in Figure 6), using only 5 random seeds raises questions about the conclusions' reliability.
> > > >
> > > > Testing the modified regression approach (q(x, s)) in the synthetic experiments, where DSO's advantages are observed, would provide more valuable insights. In such a case, the implementation of the regression model will be critical. To ensure a fair comparison, aligning the regression model with DSO by employing kernel regression using the same kernel as DSO might be a straightforward and practical choice.

---

> > > > > ### Author Response · Authors · 2024-11-26
> > > > >
> > > > > Thank you for the response. We appreciate the feedback, and we consider updating the full-LLM experiment results with more challenging configurations where the baselines fall short (e.g., higher reward noise) in the final version of the paper. We acknowledge the comments on the full-LLM experiment to be useful feedback to further strengthen our contributions.
> > > > >
> > > > > However, we consider that employing kernel regression should be more than direct applications of the existing OPL methods, and it should not be a necessary requirement for the baseline. We have several related works that propose kernel-based IS in a way different from ours [Lee et al.; 2022, Kallus and Zhou; 2018], and none of them uses kernel-based regression as the baseline (i.e., simply uses a naive regression model in the experiment).
> > > > >
> > > > > Additionally, we kindly ask the reviewer to evaluate our paper as a whole, based on the detailed theoretical and empirical analysis of when DSO provides advantages over conventional approaches. Although both regression-based PG and DSO demonstrated promising performance in the full-LLM experiment, we have already shown that DSO can achieve a suitable and steerable bias-variance tradeoff in the theoretical analysis, and demonstrated that DSO particularly works well with a large number of candidate actions and reward noises in the synthetic experiment. We appreciate your acknowledgment of these contributions in the “Strengths” section of the review.
> > > > >
> > > > > ---
> > > > >
> > > > > [Lee et al.; 2022] Haanvid Lee, Jongmin Lee, Yunseon Choi, Wonseok Jeon, Byung-Jun Lee, Yung-Kyun Noh, Kee-Eung Kim. Local Metric Learning for Off-Policy Evaluation in Contextual Bandits with Continuous Actions. NeurIPS, 2022.
> > > > >
> > > > > [Kallus and Zhou; 2018] Nathan Kallus, Angela Zhou. Policy Evaluation and Optimization with Continuous Treatments. AISTATS, 2018.

---

> > > > > > ### Comment · Reviewer_Qx8z · 2024-11-30
> > > > > >
> > > > > > Thank you for your response and for addressing some of the concerns raised. While I appreciate the effort to clarify certain points and provide additional context, several key issues remain.
> > > > > >
> > > > > > One remaining concern is the apparent lack of consideration for related work in the target domain of language modeling and applications. While the authors mainly reference foundational work in offline policy evaluation and optimization, This research is not intended to be foundational work in that area. I feel that the paper does not sufficiently engage with established approaches in NLP. (While the authors argue that kernel-based regression is unnecessary as a baseline since it is not commonly considered in OPE literature,) regression-based optimization methods using similarity functions (kernels) such as CIDEr score [1] or ROUGE [2] are, in fact, well-known and widely adopted in NLP. These methods directly correspond to the regression approach discussed in the paper and could provide a more domain-relevant baseline.
> > > > > >
> > > > > > While I recommended kernel-based regression to ensure fairness compared to DSO, its use is not mandatory. Instead, the reward predictor should be designed based on the best-known practices in the target domain or general knowledge while also considering the available tools (e.g., LLMs, data). In this context, the reward predictor should ideally depend on q(x, s), and similarity (kernel)-based regression may naturally be within the scope of consideration.
> > > > > >
> > > > > > As discussed above, another concern is their reformulation of prompt optimization, including the full-LLM experiments.
> > > > > >
> > > > > > For these reasons, I will maintain my current score.
> > > > > >
> > > > > >
> > > > > > ---
> > > > > >
> > > > > > [1] Steven J. Rennie, Etienne Marcheret, Youssef Mroueh, Jarret Ross, Vaibhava Goel. Self-critical Sequence Training for Image Captioning. CVPR, 2017
> > > > > >
> > > > > > [2] Romain Paulus, Caiming Xiong, Richard Socher. A Deep Reinforced Model for Abstractive Summarization. ICLR, 2018.
> > > > > >
> > > > > > ---
> > > > > >
> > > > > > As a supplementary note, in reinforcement learning, while textbooks often describe modeling the reward function as $r(s, a)$, it is common practice among researchers and data scientists to extend this to $r(s, a, s')$ when the task inherently depends on the next state $s'$. Similarly, in the context of this work, $q(x, s)$ may provide a more natural and informative formulation than $q(x, a)$, particularly if the generated sentences $s$ carry essential information for the reward prediction.

---

> ### Author Response · Authors · 2024-11-17
> **Responses to the review (2/4)**
>
> > **Question 1. Figure 6 Interpretation**: “The performance between DSO and regression appears similar when accounting for this variability.”
>
> We agree with the reviewer’s observation that regression also works well in the full-LLM experiments; DSO and regression perform better than other IS-based baselines. The biggest reason that regression worked well in our full-LLM experiment is that the reward noise turned out to be relatively small in the full-LLM experiment. This also aligns with the observation in the synthetic experiment, which shows that regression becomes accurate when the reward noise is zero. However, we should also note that, as we have seen in the synthetic experiment, the bias caused by regression models is often difficult to control, and similar results are often observed in many OPE/OPL papers [Swamminathan and Joachims; 2015, Metelli et al.; 2021, Saito et al.; 2021]. We will clarify these points in the revision.

---

> > ### Author Response · Authors · 2024-11-17
> > **Responses to the review (3/4)**
> >
> > Here are the responses for other comments on notations and writing.
> >
> > > **Comment 2-1. Lack of clarity in algorithmic steps**: “how many samples would need to be generated per data point? Should this match the samples used to estimate?”
> >
> > Thank you for the useful feedback. To answer the first question, we only need one sample of generated prompt and sentence per data point. Even though sampling a single data for each batch data, we can simulate the expectation (that appears in the numerator) because we repeat the process for many different batches and training steps. We also use the same sampling procedure for the regression-based approach.
> >
> > For the notation, we intended to sample one data when using the notation $z \sim p(z)$ for any random variable $z$. To improve the clarity, **we will add an explicit explanation of this in Section 3, right before the beginning of Section 3.1.**
> >
> > To answer the second question, we need $m$ samples to estimate the logging marginal density when using the Monte-Carlo estimation because it takes expectation in the denominator, unlike other expectations that are taken w.r.t. the numerator. We’d be happy to clarify this point if you have any further questions.
> >
> >
> > > **Comment 2-2. Notation abuse and lack of clarity in definitions**: “$\pi_{\theta}(a|x,s)$ .. is problematic because $\pi_{\theta}(a|x)$ is originally defined as a prompt selection policy and should not depend on $s$, which the LLM generates after selecting $a$” “$\pi_{\theta}(s|x)$ is used without explanation as $\sum_a \pi(a|x) p_{LLM}(s|x,a)$”
> >
> > Thank you for sharing your confusion around notations. We would like to kindly resolve your questions around statistics and sampling processes as follows.
> >
> > For the first point, **introducing $\pi_{\theta}(a|x,s)$ in theoretical analysis is indeed reasonable based on Bayes’ theorem.** This is because, if we consider the joint distribution of $A$ and $B$, the following holds: $P(A) P(B|A) = P(B) P(A|B)$ (where $A$ and $B$ corresponds to $a$ and $s$). The reviewer may have wondered if it is OK to use $a \sim \pi_{\theta}(a|x_i,s_i)$ in the definition of DSO, however, this is also statistically no problem, because we are sampling $a$, which is independent of $a_i$. Similar notations are also used in existing papers on OPE/OPL such as [Saito and Joachims; 2022].
> >
> > For the second point, **we have provided the definition for the marginal sentence density as $\pi_{\theta}(\phi(s)|x) = \sum_a p_{LLM}(\phi(s)|x,a) \pi(a|x)$ in the second bullet point in Section 4**. However, if the reviewer considers this to be still insufficient to denote $\pi_{\theta}(s|x)$, we’d be happy to clarify this point in the revision.
> >
> > > **Comment 2-3. Organization of the paper**: “The structure of the paper could be improved. For instance, details of the synthetic experiment setting and Section 4.2 (not cited in the main text) could be moved to the appendix, ..”
> >
> > Thank you for your thoughtful suggestions for improving the organization of the paper. However, we would like to kindly emphasize that Section 4.2 is indeed one of the central contributions of this paper and it is important to discuss the theoretical property of the proposed method in the main text (the reviewer H6Mu also acknowledges that it is our strength that “Intuition for the analysis is provided”). Nonetheless, we appreciate the reviewer’s suggestion.
> >
> > > **Minor comments**
> >
> > Thank you for pointing out typos, and we will correct them in the revision. For the classification of conventional approaches, describing "regression-based methods (also referred to as direct method)" and "importance sampling (IS)" is a common classification in the OPE/OPL literature [Aouali et al.; 2023, Hanna et al.; 2019]. However, we understand the reviewer’s viewpoint and therefore we will mention the difference between using “regressed reward” and “actual reward” in the revision.

---

> > > ### Author Response · Authors · 2024-11-17
> > > **Responses to the review (4/4)**
> > >
> > > Here we provide the references.
> > >
> > > ---
> > >
> > > [Deng et al.; 2022] Mingkai Deng, Jianyu Wang, Cheng-Ping Hsieh, Yihan Wang, Han Guo, Tianmin Shu, Meng Song, Eric P. Xing, Zhiting Hu. RLPrompt: Optimizing Discrete Text Prompts with Reinforcement Learning. EMNLP, 2022.
> > >
> > > [Saito and Joachims; 2022] Yuta Saito, Thorsten Joachims. Off-Policy Evaluation for Large Action Spaces via Embeddings. ICML, 2022.
> > >
> > > [Aouali et al.; 2023] Imad Aouali, Victor-Emmanuel Brunel, David Rohde, Anna Korba. Exponential Smoothing for Off-Policy Learning. ICML, 2023.
> > >
> > > [Hanna et al.; 2019] Josiah P. Hanna, Scott Niekum, Peter Stone. Importance Sampling Policy Evaluation with an Estimated Behavior Policy. ICML, 2019.
> > >
> > > [Swamminathan and Joachims; 2015] Adith Swaminathan, Thorsten Joachims. The Self-Normalized Estimator for Counterfactual Learning. NeurIPS, 2015.
> > >
> > > [Metelli et al., 2021] Alberto Maria Metelli, Alessio Russo, Marcello Restelli. Subgaussian and Differentiable Importance Sampling for Off-Policy Evaluation and Learning. NeurIPS, 2021.
> > >
> > > [Saito et al., 2021] Yuta Saito, Shunsuke Aihara, Megumi Matsutani, Yusuke Narita. Open Bandit Dataset and Pipeline: Towards Realistic and Reproducible Off-Policy Evaluation. NeurIPS datasets and benchmarks, 2021.

---

### Official Review · Reviewer_LAYf · 2024-11-10

**Soundness:** 2
**Presentation:** 1
**Contribution:** 2
**Rating:** 5
**Confidence:** 4

**Summary:**

This paper proposed a new policy gradient-based prompt optimization. The goal is to learn a policy that is able to generate prompts with good responses (as in good rewards). This paper proposed a new DSO that is better than the traditional policy gradient and IS-based method. Some experimental results provided by this paper show that the new method is able to outperform others.

**Strengths:**

1. The idea of learning a policy to generate good prompts is new to me.
2. The proposed method clearly addressed the weakness of IS.

**Weaknesses:**

1. The experimental session is the major weakness of this paper. This paper only contains a synthetic experiment and a single model experiment on a single dataset witha  simulated reward function. Experimental results on more datasets and models will make the paper more convincing.

2. The following work should be discussed in the related work since they study prompt optimization with human feedback by learning a reward function and hence related.

https://arxiv.org/abs/2402.00396
https://arxiv.org/abs/2405.17346

An similar line of work on prompt optimization should also be discussed:

https://arxiv.org/abs/2306.03082
https://arxiv.org/abs/2310.02905
https://arxiv.org/pdf/2402.09723

**Questions:**

1. Can the author describe the main insight for the theorems in this paper? and how they are reflected in the performance of the new approach? There seems to be some disconnection between the theoretical section and empirical verification.

2. How does your method perform in a normal prompt optimization setting? like [1]?


[1] https://arxiv.org/abs/2306.03082

---

> ### Author Response · Authors · 2024-11-17
> **Responses to the review (1/2)**
>
> We would like to thank the reviewer for valuable feedback including the suggestion of related literature. Below, we address the key comments and questions.
>
> > **Comment 1. Experiment settings.** “This paper only contain a synthetic experiment and a single model experiment on a single dataset with simulated reward function.”
>
> Thank you for the thoughtful feedback. We agree that extending the full-LLM experiment can further strengthen our paper. However, we should note that it is unfortunately quite challenging due to the **absence of existing real-world datasets that are applicable to our setting**, as the reviewer MsZw also acknowledges as “we lack good benchmarks for this task”. Indeed, one of our contributions is to enable full-LLM experiment **for the first time** by learning a realistic reward simulator from the MovieLens dataset, as acknowledged by the reviewers MsZw and H6Mu.
>
> Although many recommendation datasets are publicly available, **only the MovieLens dataset is applicable to our task due to four key qualifications**:
> 1. LLMs have knowledge about items so that they can generate sentence descriptions,
> 2. items have more than two aspects (e.g., sci-fi and romance) so that choosing a prompt makes the difference,
> 3. the effects can be different among individual contexts (i.e., users),
> 4. these differences are learnable from datasets (e.g., MovieLens enables us to learn affinity between user preference and movie features).
>
> Regarding these points, we actually did our best to replicate one of the most realistic situations from limited publicly available datasets, and demonstrated the effectiveness of our approach in a practically relevant situation.
>
> Nonetheless, we acknowledge that **publishing various real-world datasets for prompt-guided language generation could be valuable future work of the entire research community**. We will include the above discussion in the future work section (Section 8).
>
>
> > **Comment 2. Suggestions of related work on prompt optimization with human feedback.**
>
> Thank you for suggesting the related literature, and we will include them in the final version of the paper. However, **we would like to kindly note that we have already discussed similar papers and the key differences to our approach in the initial draft**.
>
> First, the suggested papers can be categorized into two sets: (1) online bandit (or online exploration) papers [Dwaracherla et al.; 2024, Chen et al.; 2023, Lin et al.; 2024, Shi et al., 2024] and (2) an RLHF paper [Lin et al.; 2024]. Because we consider the **offline** setting where we learn a policy from logged bandit data, online bandit papers are not the central related work. Therefore, we cited only the most closely related online prompt learning paper, called RLPrompt [Deng et al., 2022], as a representative and adequately discussed the key differences and limitations as “online interactions with users can be costly, harmful, or sometimes even unethical (Section 2)”. Similarly, we have a paragraph to discuss RLHF papers. The suggested paper has the same limitation of “RLHF incurs substantial cost and ethical concerns for human annotation (Section 2)” as in already cited papers.
>
> [Deng et al.; 2022] Mingkai Deng, Jianyu Wang, Cheng-Ping Hsieh, Yihan Wang, Han Guo, Tianmin Shu, Meng Song, Eric P. Xing, Zhiting Hu. RLPrompt: Optimizing Discrete Text Prompts with Reinforcement Learning. EMNLP, 2022.

---

> > ### Author Response · Authors · 2024-11-17
> > **Responses to the review (2/2)**
> >
> > > **Question 1. Can the author describe what's the main insight for the thoerms in this paper? and how they are reflected in the performance of the new approach?**
> >
> > Absolutely. The main insights from the theoretical analysis are the following three key points:
> > - **(Definition 1)** The proposed DSO is **less likely to incur deficient support**, thus can mitigate the bias issue of action-wise IS caused by missing prompts in the logged data.
> > - **(Theorem 2)** DSO **significantly reduces variance** compared to action-wise IS, and the variance reduction becomes large when the bandwidth hyperparameter is large.
> > - **(Theorem 1)** While reducing the variance, DSO also **keeps bias small by leveraging similarity among sentences via kernels**. (Because we apply IS in the marginalized sentence space, the bias is limited to the amount caused by within-kernel reward shifts, which are often small.)
> >
> > Compared to the action-wise IS that treats each prompt independently, **DSO achieves a better bias-variance tradeoff by leveraging the similarity among generated sentences**. Therefore, we can **more accurately estimate the policy gradient** via DSO, which contributes to the performance improvement as seen in the experiments.
> >
> >
> > > **Question 2. How does your method performs in normal prompt optimization setting? like [1]?**
> >
> > Thank you for raising an interesting question. Testing the usefulness of leveraging sentence similarity in the online setting would be interesting, however, we should emphasize that this is **completely out-of-scope to our paper**. Our paper solely focuses on the off-policy learning from logged data, thus we did not experiment with the online settings that existing prompt optimization papers consider. Inventing an online approach that leverages sentence similarity would be an interesting and independent future direction.

---

> > > ### Author Response · Authors · 2024-11-25
> > >
> > > Dear reviewer LAYf,
> > >
> > > Thank you once again for your valuable feedback. The author-reviewer discussion period ends soon, and we would be happy to address any further questions or concerns by then.
> > >
> > > We have also uploaded the revision including additional references based on your comments, and it would be great if you could take a look.

---

> > > > ### Comment · Reviewer_LAYf · 2024-12-02
> > > >
> > > > Thank you for your response. After reading the rebuttal, my concerns about the full-LLM evaluation (i.e., realistic datasets, more variants of models) and comparing existing prompt optimization methods still remain. Indeed, the reference I provided is online prompt optimization, however, there are still many existing prompt optimization works compared in [1] like APE [2] and OPRO [3] which use in-context learning to get the best prompt without active online learning. This paper did not consider any comparison to these existing methods, which is still a major concern. Hence I will keep my score.
> > > >
> > > > [1] https://arxiv.org/abs/2306.03082
> > > > [2] https://arxiv.org/abs/2211.01910
> > > > [3] https://arxiv.org/abs/2309.03409

---

> > > > > ### Author Response · Authors · 2024-12-02
> > > > >
> > > > > Thank you for the response. We would like to kindly point out **the reviewer’s potential misunderstanding of the term “online learning” used in our manuscript**.
> > > > >
> > > > > >  there are still many existing prompt optimization works compared in [1] like APE [2] and OPRO [3] which use in-context learning to get the best prompt without active online learning.
> > > > >
> > > > > First, what we mean by “online learning” in our manuscript is a learning approach that is applicable to a situation where *evaluation scores are easily accessible*. We clearly state this in the draft by mentioning the limitation of the existing online approach is “to assume that feedback (i.e., reward) is easily accessible” in Section 2. This applies to APE [2], which requires $f$ in Eq. (1) of [2], and OPRO [3], which requires an “objective function evaluator” as described in Figure 2 of [3]. In our baseline, the regression-based approach shares a similar strategy, as it first estimates the reward function $\hat{q}$ and exploits it during the optimization phase.
> > > > >
> > > > > Note that, in our setting, the true evaluation score function (i.e., user response function) is inaccessible, and we consider more challenging cases where user responses are logged bandit feedback, as explained in Section 3 and Figure 1.
> > > > >
> > > > > We hope this clarification resolves your concern.

---

### Author Response · Authors · 2024-11-22
**Revision**

Dear reviewers,

Thank you once again for your valuable feedback on the paper. We have uploaded the revision based on your comments. We summarize the key updates below.

- Additional discussions of related works
  - Additional papers on online learning and RLHF (Section 2, reviewer LAYf)
  - Additional discussion on kernelized bandits (Appendix A, reviewer H6Mu)
  - Additional comparison with LLM-based recommendation (Appendix A, reviewer Qx8z)
- Additional clarifications on notations (Sections 3 and 4, reviewer Qx8z)
- Additional clarification on the full-LLM experiment (Section 7, Figure 6, and Appendix C.2, reviewers H6Mu and Qx8z)
- Additional discussion of future work regarding benchmarks (Section 8, reviewer LAYf)

---

### Meta-Review · Area_Chair_1FvX · 2024-12-18

**Metareview:**

This paper proposes a method for learning a prompt policy using only offline data with bandit feedback. The proposed method is based on policy gradient and consists of specilaized techiniques to reduce both the bias and variance of the reward estimator. The paper also introduces a new benchmark for the studied problem setting of prompt optimization with offline bandit feedback.

The reviewers generally agree that the studied problem setting is interesting, and the theoretical results are insightful and useful since they provide support for practice.

The reviewers gave disparate scores for the paper even after the rebuttal. One common concern which is shared by 3 reviewers (Reviewers LAYf, Qx8z and MsZw) is that the real-LLM experiments may be insufficient, for example, the comparisons with previous methods are not enough or not entirely fair, the benchmark used in the full LLM experiments may not be representative of real scenarios. After reading the discussions between the authors and reviewers, I tend to agree with the points raised by Reviewer Qx8z. That is, the benchmark constructed using the MovieLens dataset may be too simplistic (see Reviewer Qx8z's initial review for details). I understand the authors' response stating that their method can be applied to more complex prompts, but unfortunately its performance in such experiments is not validated yet. I also agree with Reviewer Qx8z that in the regression baselines, the regression methods should take the generated sentence $s$ as input (in fact, I wonder can you let the regressors take both $a$ and $s$ as input?), and hence encourage the authors to follow this in both synthetic and full LLM experiments. The regression-based methods, including those taking $a$ and $s$ as inputs, have very close performances with the proposed DSP method in the full LLM experiments. This is another important concern since it puts into question whether the proposed method is indeed practically superior than the previous baselines. I also agree with Reviewer Qx8z's comment regarding using kernel-based regression as the regressor in the experiments for a fair comparison, since kernel similarity is also used in the proposed DSP.

I understand that the paper contains some theoretical contributions as well. However, in my opinion, for the topic studied in this paper (offline prompt optimization), having a practical algorithm (with appropriate and sufficient benchmarking) provides more value than theoretical contributions. Given the above, rejection is recommended. We encourage the authors to take into account the comments from the reviews, especially those related to the real-world LLM experiments, which will further strengthen the contributions of the paper.

**Additional Comments On Reviewer Discussion:**

During the rebuttal period, there were extensive discussions regarding the practically of the experiments and the fairness of the empirical comparisons. Unfortunately, some important concerns remain after these discussions.

---

### Decision · Program_Chairs · 2025-01-22

Reject